# HYBRID KERNEL STEIN VARIATIONAL GRADIENT DESCENT

## ABSTRACT

Stein variational gradient descent (SVGD) is a particle based approximate inference algorithm with largely well understood theoretical properties. In recent years, many variants of SVGD have been proposed and shown to share those properties. A preliminary test of the hybrid kernel variant (h-SVGD) has demonstrated promising results on image classification with deep neural network ensembles. However, the theoretical properties of h-SVGD have not yet been established, and its practical advantages have not been fully explored. In this paper, we define a hybrid kernelised Stein discrepancy (h-KSD) and prove that the h-SVGD update direction is optimal within an appropriate reproducing kernel Hilbert space. We also prove a descent lemma that guarantees a decrease in the KL divergence at each step along with other limit results. Numerical results demonstrate that h-SVGD mitigates the variance collapse behaviour of SVGD at no additional computational cost whilst remaining competitive at inference tasks.

## 1 INTRODUCTION

Approximating intractable posterior distributions is an important task in Bayesian inference and machine learning. Two widely used approaches are Markov Chain Monte Carlo (MCMC) methods and Variational Inference (VI). Despite their asymptotic guarantees, MCMC methods are often slow to converge and their samples suffer from high autocorrelation (Brooks et al., 2011). On the other hand, VI methods approximate the intractable distribution by minimising the KL divergence within a parametric family (Blei et al., 2017; Zhang et al., 2018). VI reduces the problem to one of optimisation, but the accuracy of the solution is limited by the flexibility of the parametric family.

Stein Variational Gradient Descent (SVGD) (Liu & Wang, 2016) is a deterministic sampling algorithm that imposes no parametric form on the approximating distribution, possesses asymptotic guarantees (Lu et al., 2019; Korba et al., 2020; Nüsken & Renger, 2023), and provides a fast alternative to MCMC methods (Liu et al., 2017). SVGD deterministically updates a set of particles until their empirical distribution approximates the target distribution. The update direction is that of steepest descent of the KL divergence within the zero-centred unit ball of a reproducing kernel Hilbert space (RKHS). This RKHS requires a choice of kernel, with typical choices including the radial basis function (RBF) (Liu & Wang, 2016), inverse multiquadratic (IMQ) (Gorham & Mackey, 2017), and log inverse (Chen et al., 2018) kernels.

The first term in the SVGD update direction is a kernel-smoothed gradient force that moves particles towards regions of high probability density. The second is a repulsive force between particles that prevents mode collapse. This repulsive force is driven by the same kernel used in the gradient term. The hybrid kernel variant (h-SVGD) generalises SVGD by allowing different kernels to be used in the gradient and repulsive terms. D'Angelo et al. (2021) demonstrate that in the setting of deep neural network ensembles, h-SVGD outperforms other SVGD variants on image classification tasks and improves functional diversity. However, the second kernel prevents h-SVGD from inheriting the theoretical properties of vanilla SVGD in a straightforward manner.

### 1.1 PREVIOUS WORK

The SVGD algorithm (Liu & Wang, 2016) was formulated using a connection between the kernelised Stein discrepancy (KSD) (Liu et al., 2016) and the derivative of the KL divergence. Since its

introduction, many variants have been proposed in the literature such as matrix SVGD (Wang et al., 2019), gradient free SVGD (Han & Liu, 2018), message passing or graphical SVGD (Zhuo et al., 2018; Wang et al., 2018), stochastic SVGD (Gorham et al., 2020), sliced SVGD (Gong et al., 2021), Grassman SVGD (Liu et al., 2022), and annealed SVGD (D'Angelo & Fortuin, 2021).

Liu & Wang (2016) showed that the update direction optimally decreases the KL divergence within the unit ball of the RKHS. Descent lemmas with bounds on the decrease in KL divergence have also been established (Korba et al., 2020; Salim et al., 2022; Liu, 2017).

Weak convergence in the large particle limit was demonstrated under strong assumptions (Liu, 2017) then improved with a weaker pseudo-Lipschitz assumption (Gorham et al., 2020). The empirical distribution converges in the large particle limit to a solution of a non-local nonlinear partial differential equation (PDE), and this solution converges weakly to the target distribution in the time limit (Lu et al., 2019). Korba et al. (2020) established a rate of convergence and a Wasserstein bound.

Liu (2017) showed that the dynamics of SVGD in the continuous time limit are described by the Fokker-Planck PDE. This work interpreted the PDE as a gradient flow for minimising the KL divergence functional and equated the norm of the functional gradient of the KL divergence to the KSD. The geometric interpretation has been extended in (Duncan et al., 2023; Nüsken & Renger, 2023).

The variance collapse phenomenon, in which samples generated by SVGD underestimate the variance in high dimensions, has been studied (Ba et al., 2019; 2021). Zhuo et al. (2018) provide conditions under which the size of the repulsive force correlates negatively with dimension, which leads to the gradient term dominating the SVGD dynamics and particles being prevented from representing the tails of the target distribution.

## 1.2 Contributions and outline

This paper contributes the following:

- A proof that the h-SVGD update direction is optimal within a different RKHS to that of vanilla SVGD, and a hybrid KSD (h-KSD) definition.
- A descent lemma for h-SVGD and results in the continuous time limit, the mean field limit, the hybrid kernel Stein geometry, and the finite particle regime.
- Numerical experiments demonstrating that h-SVGD can mitigate variance collapse at no extra computational cost, whilst remaining competitive at inference tasks.

Section 2 introduces notation, then recalls the necessary RKHS theory and the SVGD algorithm. The h-SVGD algorithm is presented in Section 3. In Section 4, the SVGD theory is extended to the hybrid kernel setting and the h-KSD is defined. Numerical experiments are presented in Section 5. The appendix contains proofs, a discussion on the h-KSD, and additional numerical results.

## 2 Background

### 2.1 Notation

Let $\mathcal{X} \subseteq \mathbb{R}^d$. Let $p$ denote the target density on $\mathcal{X}$, let $s_p(\boldsymbol{x}) = \nabla_{\boldsymbol{x}} \log p(\boldsymbol{x})$ denote its score function, and $\nu_p$ its measure. Assume that $p(\boldsymbol{x}) = e^{-V(\boldsymbol{x})}$ for some potential $V$. Let $\mathcal{P}(\mathcal{X})$ be the set of probability measures on $\mathcal{X}$ and $\mathcal{P}_V(\mathcal{X})$ the subset where $\|\mu\|_{\mathcal{P}_V} := \int_{\mathcal{X}} (1 + V(\boldsymbol{x})) d\mu(\boldsymbol{x}) < \infty$. For $p \geq 1$, let $\mathcal{P}_p(\mathcal{X})$ be the subset where $\|\mu\|_{\mathcal{P}_p} := \int_{\mathcal{X}} \|\boldsymbol{x}\|^p d\mu(\boldsymbol{x}) < \infty$ and define the Wasserstein $p$-distance between measures $\mu, \nu \in \mathcal{P}_p(\mathcal{X})$ as $W_p(\mu, \nu) := \left( \inf_{s \in \mathcal{S}(\mu,\nu)} \int \|\boldsymbol{x} - \boldsymbol{y}\|^p ds(\mu, \nu) \right)^{1/p}$, where $\mathcal{S}(\mu, \nu)$ is the set of couplings between $\mu$ and $\nu$. Given $\mu \in \mathcal{P}(\mathcal{X})$ and a smooth, invertible transform $\boldsymbol{T} : \mathcal{X} \to \mathcal{X}$, let $\boldsymbol{T}_{\#}\mu$ denote the pushforward measure of $\mu$ through $\boldsymbol{T}$. The KL divergence between two measures $\mu, \nu \in \mathcal{P}(\mathcal{X})$ is denoted by $\mathrm{KL}(\mu \parallel \nu)$.

## 2.2 Reproducing Kernel Hilbert Spaces

A function $k : \mathcal{X} \times \mathcal{X} \to \mathbb{R}$ is positive definite if $\sum_{i,j} a_i k(\boldsymbol{x}_i, \boldsymbol{x}_j) a_j > 0$ for any $a_1, \ldots, a_d \in \mathbb{R}$ and $\boldsymbol{x}_1, \ldots, \boldsymbol{x}_d \in \mathcal{X}$. Given a Hilbert space $\mathcal{H}$ of functions $\phi : \mathcal{X} \to \mathbb{R}$, a function $k : \mathcal{X} \times \mathcal{X} \to \mathbb{R}$ is said to be a reproducing kernel for $\mathcal{H}$ if it satisfies the reproducing property, $\phi(\boldsymbol{x}) = \langle \phi, k(\boldsymbol{x}, \cdot) \rangle_{\mathcal{H}}$ for all $\phi \in \mathcal{H}$. A positive definite $k : \mathcal{X} \times \mathcal{X} \to \mathbb{R}$ admits a unique Hilbert space $\mathcal{H}$ of functions $\phi : \mathcal{X} \to \mathbb{R}$ for which the Dirac functionals $\delta_{\boldsymbol{x}} : \mathcal{H} \to \mathbb{R}, \delta_{\boldsymbol{x}} \phi = \phi(\boldsymbol{x})$ are all continuous and $k$ is a reproducing kernel. This Hilbert space is called the reproducing kernel Hilbert space (RKHS) of $k$ and it is equal to the closure of the span of $\{k(\boldsymbol{x}, \cdot) : \boldsymbol{x} \in \mathbb{R}\}$.

Let $\mathcal{H}^d = \mathcal{H} \times \cdots \times \mathcal{H}$ denote the Hilbert space of functions $\phi : \mathcal{X} \to \mathbb{R}^d$ whose components are all in $\mathcal{H}$, and equip it with the usual inner product $\langle \phi, \psi \rangle_{\mathcal{H}^d} = \sum_{i=1}^{d} \langle \phi_i, \psi_i \rangle_{\mathcal{H}}$. Given two kernels $k_1, k_2 : \mathcal{X} \times \mathcal{X} \to \mathbb{R}$, let $\mathcal{H}_1, \mathcal{H}_2$ denote their respective RKHS. The inner product of the direct sum Hilbert space $\mathcal{H}_1^d \oplus \mathcal{H}_2^d$ is given by $\langle (\phi_1, \phi_2), (\psi_1, \psi_2) \rangle_{\mathcal{H}_1^d \oplus \mathcal{H}_2^d} = \langle \phi_1, \psi_1 \rangle_{\mathcal{H}_1^d} + \langle \phi_2, \psi_2 \rangle_{\mathcal{H}_2^d}$. The algebraic sum $\mathcal{H}_1 + \mathcal{H}_2 = \{\phi_1 + \phi_2 : \phi_1 \in \mathcal{H}_1, \phi_2 \in \mathcal{H}_2\}$ is an RKHS with kernel $k_1 + k_2$ and norm

$$\|\phi\|^2_{\mathcal{H}_1 + \mathcal{H}_2} := \min \left\{ \|\phi_1\|^2_{\mathcal{H}_1} + \|\phi_2\|^2_{\mathcal{H}_2} : \phi_1 \in \mathcal{H}_1, \phi_2 \in \mathcal{H}_2, \phi_1 + \phi_2 = \phi \right\} \tag{1}$$

for all $\phi \in \mathcal{H}_1 + \mathcal{H}_2$. It can be easily checked that $(\mathcal{H}_1 + \mathcal{H}_2)^d = \mathcal{H}_1^d + \mathcal{H}_2^d$.

For a thorough treatment of RKHS we refer the reader to (Aronszajn, 1950; Steinwart & Christmann, 2008; Berlinet & Thomas-Agnan, 2011).

## 2.3 Stein Variational Gradient Descent (SVGD)

The key result from (Liu & Wang, 2016) identifies a transform $\boldsymbol{T} : \mathcal{X} \to \mathcal{X}$ that optimally decreases the KL divergence from an arbitrary probability measure to $\nu_p$. More precisely, let $\mathcal{H}$ be an RKHS with kernel $k : \mathcal{X} \times \mathcal{X} \to \mathbb{R}$ and consider transforms of the form $\boldsymbol{T}(\boldsymbol{x}) = \boldsymbol{x} + \epsilon \phi(\boldsymbol{x})$ where $\epsilon > 0$ and $\phi$ is in the unit ball $\{\phi \in \mathcal{H}^d : \|\phi\|_{\mathcal{H}^d} \leq 1\}$. The maximum value of

$$-\nabla_\epsilon \mathrm{KL}(\boldsymbol{T}_{\#}\mu \,\|\, \nu_p)|_{\epsilon=0} \tag{2}$$

occurs at $\phi_{\mu,p}^k / \left\|\phi_{\mu,p}^k\right\|_{\mathcal{H}^d}$, where

$$\phi_{\mu,p}^k(\cdot) := \mathbb{E}_{\boldsymbol{x} \sim \mu} \left[ k(\boldsymbol{x}, \cdot) \boldsymbol{s}_p(\boldsymbol{x}) + \nabla_{\boldsymbol{x}} k(\boldsymbol{x}, \cdot) \right]. \tag{3}$$

When $\mu$ is an empirical distribution (i.e. a sum of Dirac measures), the expectation in (3) can be computed exactly by summing over the particles of each Dirac measure. Using this observation, the SVGD algorithm starts with an initial set of $N$ particles $(\boldsymbol{x}_0^i)_{i=1}^N$ and iteratively applies the transform $\boldsymbol{T}$ with (3) as the update direction. At each iteration $\ell$, this yields a set of particles $(\boldsymbol{x}_\ell^i)_{i=1}^N$ and a corresponding empirical distribution $\mu_\ell = \frac{1}{N} \sum_i \delta_{\boldsymbol{x}_\ell^i}$. This is captured in Algorithm 1. The intention is that after sufficiently many iterations, the set of particles will resemble samples from $p$ and expectations of the form $\mathbb{E}_{\boldsymbol{x} \sim p} h(\boldsymbol{x})$ can be approximated by $\mathbb{E}_{\boldsymbol{x} \sim \mu_\ell} h(\boldsymbol{x}) = \frac{1}{N} \sum_i h(\boldsymbol{x}_\ell^i)$. We also recall the definition of the KSD from (Liu et al., 2016),

$$\mathbb{S}_k(\mu, \nu_p) := \mathbb{E}_{\boldsymbol{x}, \boldsymbol{y} \sim \mu} \left[ (\boldsymbol{s}_p(\boldsymbol{x}) - \boldsymbol{s}_q(\boldsymbol{x}))^\top k(\boldsymbol{x}, \boldsymbol{y}) (\boldsymbol{s}_p(\boldsymbol{y}) - \boldsymbol{s}_q(\boldsymbol{y})) \right]. \tag{4}$$

---

**Algorithm 1** Stein Variational Gradient Descent (Liu & Wang, 2016)

---

**Input:** A target probability distribution $\nu_p$, a kernel $k$, an initial set of particles $(\boldsymbol{x}_0^i)_{i=1}^N$ in $\mathcal{X}$, and a sequence of step sizes $(\epsilon_\ell)$.
**Output:** A set of particles $(\boldsymbol{x}^i)_{i=1}^N$ in $\mathcal{X}$ whose empirical distribution approximates $\nu_p$.
**for** iteration $\ell$ **do**

$$\boldsymbol{x}_{\ell+1}^i \leftarrow \boldsymbol{x}_\ell^i + \epsilon_\ell \hat{\phi}_{\mu_\ell, p}^*(\boldsymbol{x}_\ell^i), \qquad \forall \ i = 1, \ldots, N \tag{5}$$

$$\hat{\phi}_{\mu_\ell, p}^*(\boldsymbol{x}) = \frac{1}{N} \sum_{j=1}^N \left( k(\boldsymbol{x}_\ell^j, \boldsymbol{x}) \nabla_{\boldsymbol{x}_\ell^j} \log p(\boldsymbol{x}_\ell^j) + \nabla_{\boldsymbol{x}_\ell^j} k(\boldsymbol{x}_\ell^j, \boldsymbol{x}) \right)$$

---

## 3   Hybrid Kernel Stein Variational Gradient Descent (h-SVGD)

The SVGD update in (5) contains two terms, each using the same kernel. The driving term uses the score function to move particles towards regions of high probability density, and the repulsive term prevents particles from collapsing at the modes. The h-SVGD variant proposed by D'Angelo et al. (2021) uses a different kernel in each term. Let $k_1$ denote the kernel that appears alongside the score function, let $k_2$ denote the repulsive kernel, and define $\Delta k := k_2 - k_1$. For the remainder of this paper, $k_1$ and $k_2$ will both be positive definite. We present h-SVGD in Algorithm 2.

---

**Algorithm 2** Hybrid Kernel Stein Variational Gradient Descent

---

**Input:** A target probability distribution $\nu_p$, two kernels $k_1$, $k_2$, an initial set of particles $(\boldsymbol{x}_0^i)_{i=1}^N$ in $\mathcal{X}$, and a sequence of step sizes $(\epsilon_\ell)$.
**Output:** A set of particles $(\boldsymbol{x}^i)_{i=1}^N$ in $\mathcal{X}$ whose empirical distirbution approximates $\nu_p$.
**for** iteration $\ell$ **do**

$$\boldsymbol{x}_{\ell+1}^i \leftarrow \boldsymbol{x}_\ell^i + \epsilon_\ell \hat{\phi}_{\mu_\ell,p}^*(\boldsymbol{x}_\ell^i), \qquad \forall i = 1, \dots, N \quad (6)$$

$$\hat{\phi}_{\mu_\ell,p}^*(\boldsymbol{x}) = \frac{1}{N} \sum_{j=1}^N \left( k_1(\boldsymbol{x}_\ell^j, \boldsymbol{x}) \nabla_{\boldsymbol{x}_\ell^j} \log p(\boldsymbol{x}_\ell^j) + \nabla_{\boldsymbol{x}_\ell^j} k_2(\boldsymbol{x}_\ell^j, \boldsymbol{x}) \right)$$

---

## 4   Theoretical Properties of h-SVGD

In this section, we identify a natural RKHS in which to develop the h-SVGD theory and justify its legitimacy as a SVGD variant. We prove that the h-SVGD update direction is optimal within this RKHS and we define a h-KSD. Other results include a descent lemma, convergence results, a gradient flow interpretation, and a Wasserstein bound. All proofs are provided in the appendix.

### 4.1   Definitions and Assumptions

A function $f : \mathcal{X} \to \mathbb{R}$ is in the Stein class of $p$ (or its corresponding measure) if it is smooth and $\int_{\boldsymbol{x} \in \mathcal{X}} \nabla_{\boldsymbol{x}} (f(\boldsymbol{x}) p(\boldsymbol{x})) \, d\boldsymbol{x} = 0$. A function $\boldsymbol{f} = (f_1, \dots, f_d) : \mathcal{X} \to \mathbb{R}^d$ is in the Stein class of $p$ if each $f_i$ is in the Stein class of $p$. A kernel $k : \mathcal{X} \times \mathcal{X} \to \mathbb{R}$ is in the Stein class of $p$ if it has continuous second order partial derivatives and both $k(\boldsymbol{x}, \cdot)$ and $k(\cdot, \boldsymbol{y})$ are in the Stein class of $p$. The hybrid Stein operator of $p$ may act on a pair of scalar functions $f, g : \mathcal{X} \to \mathbb{R}$, a pair of vector functions $\boldsymbol{f}, \boldsymbol{g} : \mathcal{X} \to \mathbb{R}^d$, or a pair of kernels $k_1, k_2 : \mathcal{X} \times \mathcal{X} \to \mathbb{R}$ by

$$\mathcal{S}_p(f, g)(\boldsymbol{x}) := f(\boldsymbol{x}) \boldsymbol{s}_p(\boldsymbol{x}) + \nabla_{\boldsymbol{x}} g(\boldsymbol{x}),$$
$$\mathcal{S}_p(\boldsymbol{f}, \boldsymbol{g})(\boldsymbol{x}) := \boldsymbol{s}_p(\boldsymbol{x})^\top \boldsymbol{f}(\boldsymbol{x}) + \nabla_{\boldsymbol{x}} \boldsymbol{g}(\boldsymbol{x}),$$
$$\mathcal{S}_p \otimes (k_1, k_2)(\boldsymbol{x}, \cdot) := k_1(\boldsymbol{x}, \cdot) \boldsymbol{s}_p(\boldsymbol{x}) + \nabla_{\boldsymbol{x}} k_2(\boldsymbol{x}, \cdot),$$

respectively. This reduces to the Stein operator (Liu et al., 2016) when acting on two equal functions.

The results in this section require the following technical assumptions on the potential function.

(**A1**) $V \in C^\infty(\mathcal{X})$, $V \geq 0$, and $\lim_{|\boldsymbol{x}| \to \infty} V(\boldsymbol{x}) = 0$.

(**A2**) There exist constants $C_V > 0$ and $q > 1$ such that

$$|V(\boldsymbol{x})|^q \leq C_V(1 + V(\boldsymbol{x}))$$

for all $\boldsymbol{x} \in \mathcal{X}$, and

$$\sup_{\theta \in [0,1]} \left| \nabla^2 V(\theta \boldsymbol{x} + (1-\theta)\boldsymbol{y}) \right|^q \leq C_V(1 + V(\boldsymbol{x}) + V(\boldsymbol{y})).$$

(**A3**) For any $\alpha, \beta > 0$, there exists a constant $C_{\alpha,\beta} > 0$ such that

$$|\boldsymbol{y}| \leq \alpha |\boldsymbol{x}| + \beta \implies (1 + |\boldsymbol{x}|)(|\nabla V(\boldsymbol{y})| + \left| \nabla^2 V(\boldsymbol{y}) \right|) \leq C_{\alpha,\beta}(1 + V(\boldsymbol{x})).$$

(**A4**) The Hessian $H_V$ of $V$ is well-defined and $\|H_V\|_{\mathrm{op}} \leq M$ for some $M > 0$.

Assumptions (**A1**), (**A2**) and (**A3**) are identical to those in (Lu et al., 2019), and Assumption (**A4**) is identical to Assumption (A2) in (Korba et al., 2020). Assumptions on the kernels are also required.

(**B1**) There exist symmetric functions $K_1, K_2 : \mathcal{X} \to \mathbb{R}$ such that $k_1(\boldsymbol{x}, \boldsymbol{y}) = K_1(\boldsymbol{x} - \boldsymbol{y})$, $k_2(\boldsymbol{x}, \boldsymbol{y}) = K_2(\boldsymbol{x} - \boldsymbol{y})$, $K_1$ is $C^2$ with bounded derivatives, and $K_2$ is $C^4$ with bounded derivatives. We use $B$ as a bound for all derivatives in the proofs.

(**B2**) There exists a constant $D > 0$ such that both $k_1$ and $\nabla k_2$ and are $D$-Lipschitz, and $\nabla V(\cdot)k_1(\cdot, \boldsymbol{z})$ is $D$-Lipschitz for each $\boldsymbol{z}$. That is,
$$|k_1(\boldsymbol{x}, \boldsymbol{x}') - k_1(\boldsymbol{y}, \boldsymbol{y}')| \leq D\left(\|\boldsymbol{x} - \boldsymbol{y}\|_2 + \|\boldsymbol{x}' - \boldsymbol{y}'\|_2\right),$$
$$\|\nabla_{\boldsymbol{x}}k_2(\boldsymbol{x}, \boldsymbol{x}') - \nabla_{\boldsymbol{y}}k_2(\boldsymbol{y}, \boldsymbol{y}')\| \leq D\left(\|\boldsymbol{x} - \boldsymbol{y}\|_2 + \|\boldsymbol{x}' - \boldsymbol{y}'\|_2\right),$$
$$|\nabla V(\boldsymbol{x})k_1(\boldsymbol{x}, \boldsymbol{z}) - \nabla V(\boldsymbol{y})k_1(\boldsymbol{y}, \boldsymbol{z})| \leq D\left(\|\boldsymbol{x} - \boldsymbol{y}\|_2\right)$$
for all $\boldsymbol{x}, \boldsymbol{x}', \boldsymbol{y}, \boldsymbol{y}', \boldsymbol{z} \in \mathcal{X}$.

(**B3**) The function $\mathcal{S}_p \otimes (k_1, k_2)(\cdot, \cdot)$ is jointly pseudo-Lipschitz. That is, there exists a constant $L > 0$ such that
$$\|\mathcal{S}_p \otimes (k_1, k_2)(\boldsymbol{x}, \boldsymbol{y}) - \mathcal{S}_p \otimes (k_1, k_2)(\boldsymbol{x}', \boldsymbol{y})\|_2 \leq L\left(1 + \|\boldsymbol{y}\|_2\right)\|\boldsymbol{x} - \boldsymbol{x}'\|_2$$
$$\|\mathcal{S}_p \otimes (k_1, k_2)(\boldsymbol{x}, \boldsymbol{y}) - \mathcal{S}_p \otimes (k_1, k_2)(\boldsymbol{x}, \boldsymbol{y}')\|_2 \leq L\left(1 + \|\boldsymbol{x}\|_2\right)\|\boldsymbol{y} - \boldsymbol{y}'\|_2$$
for all $\boldsymbol{x}, \boldsymbol{x}', \boldsymbol{y}, \boldsymbol{y}' \in \mathcal{X}$.

Assumption (**B1**) is a slight relaxation of Assumption 2.1 in (Lu et al., 2019). The first two parts of Assumption (**B2**) are hybrid kernel versions of Assumption (B2) from (Korba et al., 2020). The third part of Assumption (**B2**) replaces the restrictive Assumption (B1) from (Korba et al., 2020). The single kernel version of Assumption (**B3**) was introduced in (Gorham et al., 2020, Theorem 7).

## 4.2 Update Direction and h-KSD

Motivated by the h-SVGD update in (6), define the update direction as
$$\phi_{\mu,p}^{k_1,k_2}(\cdot) := \mathbb{E}_{\boldsymbol{x} \sim \mu}\left[\mathcal{S}_p \otimes (k_1, k_2)(\cdot, \boldsymbol{x})\right]. \tag{7}$$
Let $\boldsymbol{G}(\,\cdot\,; k_1, \mu, p) := \mathbb{E}_{\boldsymbol{x} \sim \mu}\left[k_1(\boldsymbol{x}, \cdot)\boldsymbol{s}_p(\boldsymbol{x})\right]$ and $\boldsymbol{R}(\,\cdot\,; k_2, \mu) := \mathbb{E}_{\boldsymbol{x} \sim \mu}\left[\nabla_{\boldsymbol{x}}k_2(\boldsymbol{x}, \cdot)\right]$ be the gradient and repulsive terms respectively, so $\phi_{\mu,p}^{k_1,k_2}(\cdot) = \boldsymbol{G}(\cdot; k_1, \mu, p) + \boldsymbol{R}(\cdot; k_2, \mu)$. The update transform
$$\boldsymbol{T}_{\mu,p}^{k_1,k_2}(\boldsymbol{x}) = \boldsymbol{x} + \epsilon\phi_{\mu,p}^{k_1,k_2}(\boldsymbol{x}) \tag{8}$$
and the map $\Phi_p^{k_1,k_2} : \mu \mapsto (\boldsymbol{T}_{\mu,p}^{k_1,k_2})_{\#}\mu$ characterise the h-SVGD dynamics. For each $\ell$, define
$$\mu_{\ell+1}^N := \Phi_p^{k_1,k_2}(\mu_\ell^N), \qquad\qquad \mu_{\ell+1}^\infty := \Phi_p^{k_1,k_2}(\mu_\ell^\infty), \tag{9}$$
where $\mu_0^N$ is the empirical measure of the initial particles $(\boldsymbol{x}_0^i)_{i=1}^N$ drawn i.i.d. from some $\mu_0^\infty$.

**Remark 1.** *If $k_2$ is in the Stein class of $\mu$, then $\boldsymbol{G}$ and $\boldsymbol{R}$ can be written in terms of Hilbert-Schmidt integral operators with kernels $k_1$ and $k_2$ respectively. Since those operators map into $\mathcal{H}_1$ and $\mathcal{H}_2$ (Steinwart & Christmann, 2008, Theorem 4.26), we have $\boldsymbol{G} \in \mathcal{H}_1^d$ and $\boldsymbol{R} \in \mathcal{H}_2^d$, provided that $\log p, \log q \in L_2(\mu)$. This suggests that we optimise the update direction within $\mathcal{H}_1^d + \mathcal{H}_2^d$.*

The following result justifies the definition of the update direction (7). When $\mathcal{H}_1 = \mathcal{H}_2$, it reduces to Lemma 3.2 in (Liu & Wang, 2016) up to scaling.

**Theorem 4.1.** *Suppose that $k_1$ and $k_2$ are in the Stein class of p. Then the function from the set $\mathcal{B} = \{\phi \in \mathcal{H}_1^d \cap \mathcal{H}_2^d : \|2\phi\|_{\mathcal{H}_1^d + \mathcal{H}_2^d} \leq 1\}$ that maximises (2) is $\phi_{\mu,p}^{k_1,k_2} / \left\|2\phi_{\mu,p}^{k_1,k_2}\right\|_{\mathcal{H}_1^d + \mathcal{H}_2^d}$, and this maximum value is $\left\|\phi_{\mu,p}^{k_1,k_2}\right\|_{\mathcal{H}_1^d + \mathcal{H}_2^d}$.*

Since it is known that $-\nabla_\epsilon \mathrm{KL}(\boldsymbol{T}_{\#}\mu \,\|\, \nu_p)|_{\epsilon=0} = \mathbb{E}_{\boldsymbol{x} \sim \mu}\left[\mathcal{S}_p\phi(\boldsymbol{x})\right]$ (Liu & Wang, 2016, Theorem 3.1), the above result motivates the definition of a hybrid kernelised Stein discrepancy (h-KSD) given by
$$\mathbb{S}_{k_1,k_2}(\mu, \nu_p) := \max_{\phi \in \mathcal{H}_1^d \cap \mathcal{H}_2^d}\left\{\mathbb{E}_{\boldsymbol{x} \sim \mu}\left[\mathcal{S}_p\phi(\boldsymbol{x})\right]^2 : \|2\phi\|_{\mathcal{H}_1^d + \mathcal{H}_2^d} \leq 1\right\}. \tag{10}$$

With this definition, and noting that $\mathbb{S}_{k_1,k_2}(\mu, \nu_p) = \left\|\phi_{\mu,p}^{k_1,k_2}\right\|_{\mathcal{H}_1^d + \mathcal{H}_2^d}^2$, Theorem 4.1 generalises Theorem 3.8 from (Liu et al., 2016). Note also that $\mathcal{H}_1^d \cap \mathcal{H}_2^d \subseteq \mathcal{H}_1^d + \mathcal{H}_2^d$.

**Remark 2.** *The inclusion $\mathcal{H}_1 \subseteq \mathcal{H}_2$ is equivalent to $k_1 \leq ck_2$ for some constant $c > 0$ (Aronszajn, 1950, Part I, Section 13). This is easily verified for many typical kernel choices, including when $k_1$ and $k_2$ are any combination of RBF, IMQ, log-inverse, or Matérn kernels of any bandwidth. So for kernels of these forms, the optimisation of Theorem 4.1 takes place within $\mathcal{H}_1^d$ or $\mathcal{H}_2^d$.*

Note that Liu et al. (2016) defines the KSD using the score difference. They also present a tractable formula and a spectral decomposition. Appendix C details why those three forms do not reconcile with our variational definition of the h-KSD. An interesting piece of future work is to find a tractable h-KSD formula and apply it to problems such as the goodness-of-fit tests in (Liu et al., 2016).

### 4.3 LARGE TIME ASYMPTOTICS

The following result extends Theorem 3.3 of (Liu, 2017) to the hybrid kernel setting. It shows that for a sufficiently small step size, h-SVGD will always decrease the KL divergence. This type of result is referred to as a descent lemma in the literature. Alternative descent lemmas for SVGD may be found in (Korba et al., 2020, Proposition 5) and (Salim et al., 2022, Theorem 3.2). We remark that although the h-KSD is not a valid discrepancy measure, as previously mentioned, the following descent lemma bounds the decrease in KL divergence in terms of the KSD of one of the individual kernels. This ensures the KL divergence is strictly decreasing at all times, meaning that the h-SVGD algorithm avoids the case where $\mathbb{S}_{k_1,k_2}(\mu_\ell^\infty, \nu_p) = 0$ but $\mu_\ell^\infty$ and $\nu_p$ are not equal almost everywhere.

**Theorem 4.2.** *Set $\epsilon_\ell \leq (2\sup_x \rho(\nabla\phi^* + \nabla\phi^{*\top}))^{-1}$ where $\rho(\cdot)$ denotes the spectrum radius of a matrix and $\phi^*$ is the optimum direction from Theorem 4.1. Then there exist constants $C_1, C_2 > 0$ such that $\mathcal{H}_2 \subseteq \mathcal{H}_1$ implies*

$$\mathrm{KL}(\mu_{\ell+1}^\infty \parallel \nu_p) - \mathrm{KL}(\mu_\ell^\infty \parallel \nu_p) \leq -\epsilon_\ell \mathbb{S}_{k_1}(\mu_\ell, \nu_p) + \epsilon_\ell^2 \left( C_1 \left( \mathbb{S}_{k_1}(\mu_\ell, \nu_p) + \|R_1\|_{\mathcal{H}_1^d}^2 \right) + C_2 \right)$$

*and $\mathcal{H}_1 \subseteq \mathcal{H}_2$ implies*

$$\mathrm{KL}(\mu_{\ell+1}^\infty \parallel \nu_p) - \mathrm{KL}(\mu_\ell^\infty \parallel \nu_p) \leq -\epsilon_\ell \mathbb{S}_{k_2}(\mu_\ell, \nu_p) + \epsilon_\ell^2 \left( C_1 \left( \mathbb{S}_{k_2}(\mu_\ell, \nu_p) + \|R_2\|_{\mathcal{H}_2^d}^2 \right) + C_2 \right)$$

*where we define $R_1(\boldsymbol{x}) := \mathbb{E}_{\boldsymbol{y}\sim\mu_\ell}[\nabla_{\boldsymbol{y}}\Delta k(\boldsymbol{y}, \boldsymbol{x})]$ and $R_2(\boldsymbol{x}) := \mathbb{E}_{\boldsymbol{y}\sim\mu_\ell}[\Delta k(\boldsymbol{y}, \boldsymbol{x})\boldsymbol{s}_p(\boldsymbol{y})]$. The constant $C_1$ depends only on $p$, $k_1$ and $k_2$. However, $C_2$ depends on $p$, $k_1$, $k_2$ and $\mu_\ell$.*

It can be easily verified that in the $\mathcal{H}_2 \subseteq \mathcal{H}_1$ case, the right hand side is negative when

$$\epsilon_\ell < \frac{1}{C_1} \left( \frac{S_{k_1}(\mu_\ell, \nu_p)}{S_{k_1}(\mu_\ell, \nu_p) + \|R_1\|_{\mathcal{H}_1^d}^2 + \frac{C_2}{C_1}} \right),$$

with a similar condition in the $\mathcal{H}_1 \subseteq \mathcal{H}_2$ case. Although it is not explicit in (Liu, 2017), vanilla SVGD requires a similar condition to ensure the KL divergence decreases at each step.

In Section 5 we show that a stronger repulsive kernel, that is $k_1 \leq k_2$, can mitigate variance collapse. In this scheme, $\mathcal{H}_1 \subseteq \mathcal{H}_2$ by Remark 2, and so the rate of decrease of the KL divergence is dominated by $\mathbb{S}_{k_2}$. Loosely speaking, we have $\mathbb{S}_{k_2} \leq \mathbb{S}_{k_1}$ in this case (see Figure 3 of Appendix C), so h-SVGD can mitigate variance collapse at the expense of slower convergence.

### 4.4 LARGE PARTICLE LIMIT

The following result extends weak convergence in the population limit (Gorham et al., 2020, Theorem 7) to the hybrid kernel setting. The only modification is that we require the pseudo Lipschitz assumption to apply to the hybrid Stein operator instead of the single kernel Stein operator.

**Proposition 4.3.** *If $W_1(\mu_0^N, \mu_0^\infty) \to 0$ and Assumption (B3) holds, then $W_1(\mu_\ell^N, \mu_\ell^\infty) \to 0$ at each iteration $\ell$.*

### 4.5 CONTINUOUS TIME LIMIT AND GRADIENT FLOW

In this subsection, we recall some definitions from (Liu, 2017) and restate results that apply in the space $\mathcal{H} := \mathcal{H}_1^d + \mathcal{H}_2^d$. The derivation of the Fokker-Planck equation follows without modification,

$$\frac{\partial}{\partial t} q_t(\boldsymbol{x}) = -\nabla \cdot \left( \phi_{\mu_t, p}^{k_1, k_2}(\boldsymbol{x}) q_t(\boldsymbol{x}) \right),$$

where $\mu_t$ are measures corresponding to densities $q_t$ in the continuous time $t$. In this continuous time limit, we can extend Theorem 3.4 of (Liu, 2017) to the hybrid kernel setting. We remark that this is a continuous version of Theorem 4.2 and gives a rate of convergence for the h-SVGD dynamics.

**Proposition 4.4.** *Assume* $\mathrm{KL}(\mu_0 \parallel \nu_p) < \infty$. *If* $\mathcal{H}_2 \subseteq \mathcal{H}_1$, *then* $\frac{d}{dt}\mathrm{KL}(\mu_t \parallel \nu_p) = -\mathbb{S}_{k_1}(\mu_t, \nu_p)$. *Similarly, if* $\mathcal{H}_1 \subseteq \mathcal{H}_2$, *then* $\frac{d}{dt}\mathrm{KL}(\mu_t \parallel \nu_p) = -\mathbb{S}_{k_2}(\mu_t, \nu_p)$.

Recall that the spaces $\mathcal{H}_q := \{\mathcal{S}_q\phi : \phi \in \mathcal{H}_1^d \cap \mathcal{H}_2^d\}$ and $q\mathcal{H}_q := \{q\mathcal{S}_q\phi : \phi \in \mathcal{H}_1^d \cap \mathcal{H}_2^d\}$ are equipped with the inner products $\langle f, g \rangle_{\mathcal{H}_q} := \langle qf, qg \rangle_{q\mathcal{H}_q} := \langle \psi_{q,f}, \psi_{q,g} \rangle_{\mathcal{H}}$, where

$$\psi_{q,f} := \underset{\psi \in \mathcal{H}_1^d \cap \mathcal{H}_2^d}{\arg\min} \{\|\psi\|_{\mathcal{H}} : \mathcal{S}_q\psi = f\}$$

for each $f \in \mathcal{H}_q$. Recall the covariant gradient of the functional $F$ is an element in the tangent space $q\mathcal{H}_q$ satisfying $F(q+fdt) = F(q) + \langle \mathrm{grad}_{\mathcal{H}}F(q), fdt \rangle_{q\mathcal{H}_q}$ for all $f \in q\mathcal{H}_q$. We now generalise part of Theorem 3.5 from (Liu, 2017) as a first step towards understanding the hybrid Stein geometry.

**Proposition 4.5.** *In the space* $\mathcal{H} = \mathcal{H}_1^d + \mathcal{H}_2^d$,

$$\mathrm{grad}_{\mathcal{H}}\mathrm{KL}(\mu \parallel \nu_p) = \nabla \cdot (\phi_{q,p}^{k_1,k_2}q), \text{ and} \qquad \frac{\partial q_t}{\partial t} = -\mathrm{grad}_{\mathcal{H}}\mathrm{KL}(\mu_t \parallel \nu_p).$$

It is not clear whether $\|\mathrm{grad}_{\mathcal{H}}\mathrm{KL}(\mu \parallel \nu_p)\|_{q\mathcal{H}_q}^2 = \mathbb{S}_k(\mu, \nu_p)$ from (Liu, 2017, Theorem 3.5) can be extended to the hybrid kernel setting due to issues with the h-KSD (see Appendix C). We leave a detailed study of the hybrid Stein geometry and its connection to the h-KSD for future work.

### 4.6 MEAN FIELD PDE

This subsection reviews the setup of (Lu et al., 2019) and borrows some notation from (Duncan et al., 2023). We write $K_i(\boldsymbol{x} - \boldsymbol{y})$ in place of $k_i(\boldsymbol{x}, \boldsymbol{y})$ for $i = 1, 2$ (as per Assumption (**B**1)), noting that $-\nabla K_i(\boldsymbol{x} - \boldsymbol{y}) = \nabla_{\boldsymbol{y}}k_i(\boldsymbol{x}, \boldsymbol{y})$. Also, recall that the density $p(\boldsymbol{x})$ is assumed to take the form $e^{-V(\boldsymbol{x})}$ for some potential function $V : \mathcal{X} \to \mathbb{R}$, so that $\boldsymbol{s}_p(\boldsymbol{x}) = -\nabla_{\boldsymbol{x}}V(\boldsymbol{x})$. In the continuous time limit, the evolution in Algorithm 2 can be described by the following interacting particle system,

$$\frac{d\boldsymbol{x}_t^i}{dt} = \frac{1}{N}\sum_{j=1}^{N}\left(-K_1(\boldsymbol{x}_t^i - \boldsymbol{x}_t^j)V(\boldsymbol{x}_t^j) - \nabla_{\boldsymbol{x}_t^j}K_2(\boldsymbol{x}_t^i - \boldsymbol{x}_t^j)\right). \tag{11}$$

In the mean field limit, this particle system can be described by the following PDE,

$$\partial_t\mu_t(\boldsymbol{x}) = \nabla_{\boldsymbol{x}} \cdot \left(\mu_t(\boldsymbol{x})\int_{\mathcal{X}}\left[K_1(\boldsymbol{x} - \boldsymbol{y})\nabla V(\boldsymbol{y}) - \nabla_{\boldsymbol{y}}K_2(\boldsymbol{x} - \boldsymbol{y})\right]\mu_t(d\boldsymbol{y})\right). \tag{12}$$

**Definition 4.1.** *Given a probability measure* $\nu$ *on* $\mathbb{R}^d$, *a map* $X(t, \boldsymbol{x}; \nu) : [0, \infty) \times \mathbb{R}^d \to \mathbb{R}^d$ *that is* $C^1$ *with respect to* $t$ *and satisfies*

$$\begin{aligned}\partial_t X(t, \boldsymbol{x}; \nu) &= -(K_1 * (\nabla V\mu_t))(X(t, \boldsymbol{x}; \nu)) - (K_1 * \mu_t)(X(t, \boldsymbol{x}; \nu)) \\ \mu_t &= X(t, \cdot, \nu)_{\#}\nu \\ X(0, \boldsymbol{x}; \nu) &= \boldsymbol{x}.\end{aligned} \tag{13}$$

*is called a mean field characteristic flow of (11) or of (12)*

We introduce some notation here before generalising Theorem 2.4 from (Lu et al., 2019). The set $Y := \{u \in C(\mathcal{X}, \mathcal{X}) : \sup_{\boldsymbol{x} \in \mathcal{X}}|u(\boldsymbol{x}) - \boldsymbol{x}| < \infty\}$ with $d_Y(u, v) = \sup_{\boldsymbol{x} \in \mathcal{X}}|u(\boldsymbol{x}) - v(\boldsymbol{x})|$ is a complete metric space.

**Proposition 4.6.** *Assume (A1) and (**B**1), and let* $T > 0$. *Then there exists a unique solution* $X(\cdot, \cdot, \nu) \in C^1([0, T]; Y)$ *to (13) and the corresponding* $\mu_t$ *is a weak solution to (12) that satisfies*

$$\|\mu_t\|_{\mathcal{P}_V} \leq \|\nu_p\|_{\mathcal{P}_V}\exp\left(C\min\left(\|\nabla K_1\|_{\infty}, \|\nabla K_2\|_{\infty}\right)t\right)$$

*for some constant* $C > 0$ *depending on* $K_1$, $K_2$ *and* $V$.

The second kernel enables a stronger bound by careful modification of the proof (see Appendix A). We remark that this bound describes regularity of the solution to the PDE, not a rate of convergence.

### 4.7 FINITE PARTICLES REGIME

**Proposition 4.7.** *Assume (**A1**), (**A4**), (**B1**) and (**B2**), and let $T > 0$. For any $0 \leq \ell \leq \frac{T}{\epsilon_\ell}$, there exists a constant $L$ depending on $k_1, k_2$ and $p$ such that*

$$\mathbb{E}\left[W_2^2(\mu_\ell^N, \mu_\ell^\infty)\right] \leq \frac{1}{2\sqrt{N}}\sqrt{\mathrm{var}(\mu_0^\infty)}e^{LT}(e^{2LT} - 1).$$

The single kernel version of this result (Korba et al., 2020, Proposition 7) uses an assumption that is quite restrictive. It requires $|V(\boldsymbol{x})| \leq C_V$ for some constant $C_V > 0$, which rules out even a normal target distribution. We relax this with the third part of Assumption (**B2**) and provide an updated proof in the appendix along with a hybrid kernel version of (Korba et al., 2020, Lemma 14).

## 5 EXPERIMENTS

The problem of variance collapse in SVGD has been successfully prevented in the setting of probabilistic graphical models where the conditional dependence structure enables $p$ to be factorised and $\boldsymbol{R}(\,\cdot\,; k_2, \mu)$ to be replaced with a set of lower dimensional repulsive forces (Zhuo et al., 2018). Other methods such as S-SVGD (Gong et al., 2021) and GSVGD (Liu et al., 2022) have demonstrated that variance collapse can be avoided, although there is an additional computational cost required in these methods. In particular, S-SVGD requires computation of the optimal test directions, and GSVGD requires the projectors to be updated at each step. Our numerical experiments demonstrate that even without a conditional dependence structure, and without incurring additional computational cost, h-SVGD can mitigate variance collapse while maintaining or improving the inference capabilities of SVGD. We emphasise that the cost of SGVD and h-SVGD updates are both $O(N^2)$. We measure variance collapse using dimension averaged marginal variance (DAMV), $\frac{1}{d}\sum_{j=1}^{d}\mathrm{Var}_j\left(\{\boldsymbol{x}_i\}_{i=1}^N\right)$, as is standard in the literature (Ba et al., 2019; 2021; Zhuo et al., 2018).

Zhuo et al. (2018) attribute variance collapse to the negative correlation between $\|\boldsymbol{R}(\,\cdot\,; k_2, \mu)\|_\infty$ and the dimension $d$. The intuition is that a weak repulsive force in high dimensions will enable the driving force to move particles closer to a mode, leaving the tails of the distribution underrepresented. Our numerical experiments use an RBF kernel for $k_1$ with the bandwidth set using the median particle distance heuristic (Liu & Wang, 2016). We implement a stronger repulsive kernel $k_2 = f(d)k_1$, where $f$ is an increasing function. The vanilla SVGD, $k_1 = k_2$, is presented as a baseline in each experiment. We emphasise that as per Remark 2, all of our experiments satisfy the conditions of Theorem 4.2. See Appendix D for further experimental details and results.

### 5.1 VARIANCE COLLAPSE IN THE PROPORTIONAL LIMIT

In this first example, we run h-SVGD on a sequence of multivariate normal (MVN) distributions up to a dimension $d = 200$ with marginal moments of lower dimensions equal across each MVN distributions. Following (Ba et al., 2021), the number of particles $N$ is chosen such that $d/N$ approaches a proportional limit $\gamma$. We present results in three schemes, $\gamma < 1$, $\gamma = 1$ and $\gamma > 1$. Figure 1 shows that scaling the repulsive kernel by $\sqrt{d}$ or $\ln(d)$ mitigates variance collapse and even slightly improves the mean estimates in higher dimensions, even in the absence of a conditional dependence structure. The quality of the mean estimate is measured by the dimension averaged squared mean error (DASME), $\frac{1}{d}\sum_{j=1}^{d}\left(\mathbb{E}_j\left[\{\boldsymbol{x}_i\}_{i=1}^N\right] - \mu_j\right)^2$ where $\mu_{d,j}$ is the $j$-th marginal mean of the $d$-dimensional MVN.

### 5.2 BAYESIAN NEURAL NETWORK

In this section, we sample weights from a Bayesian neural network. Aside from scaling the repulsive kernel, our setup is identical to Liu & Wang (2016). Figure 2 shows that the problem of variance collapse is mitigated under h-SVGD whilst remaining competitive in inference capabilities. This is demonstrated by an improved DAMV when scaling the repulsive kernel by $\sqrt{d}$, while the test metrics of RMSE and LL remain within the range of standard error of the SVGD experiments, with the exception of the Yacht dataset. Similar results can be seen when scaling the repulsive kernel by $\log(d)$. The values used in Figure 2 are reported in Appendix D (see Tables 1-10). We leave a more

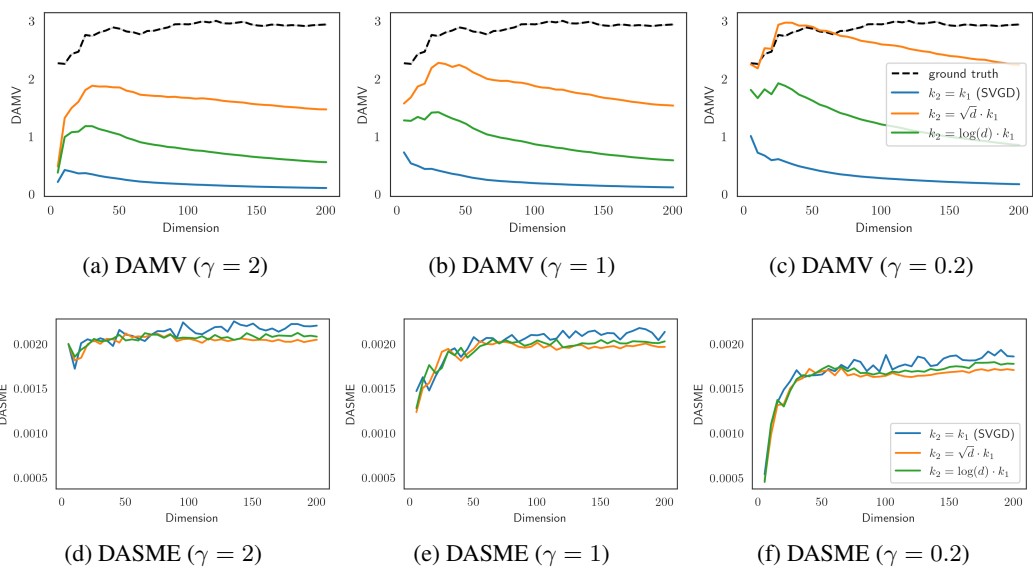

Figure 1: DAMV and DASME of MVN distributions in the proportional limit with kernel scaling.

detailed study of the scaling factor as an area for future work. Appendix D also contains additional results for this experiment with other kernel forms.

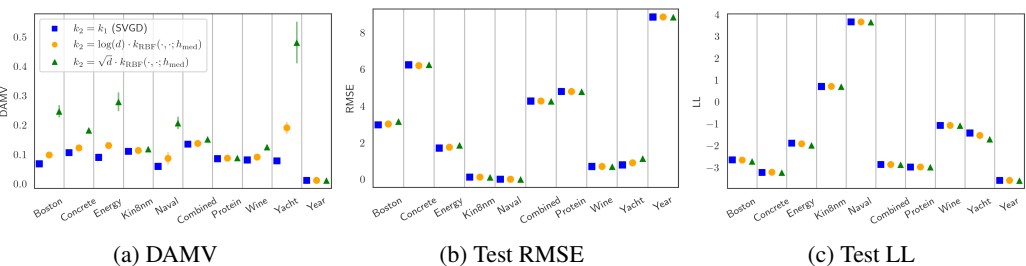

Figure 2: DAMV, RMSE and LL metrics with standard errors for two different scaling choices.

# 6 CONCLUSION

We developed the theory of the h-SVGD algorithm and defined a h-KSD. These results guarantee that h-SVGD decreases the KL divergence at each iteration and provide convergence guarantees in several limits. Our experimental results demonstrate that h-SVGD with a stronger repulsive kernel mitigates variance collapse whilst remaining competitive in standard inference tasks. We hypothesise that the mitigation of variance collapse comes at the expense of a slower convergence rate and leave a rigorous verification of this for future work. Two other promising directions for future work in the h-SVGD theory are reconciling the variational definition of the h-KSD with other explicit forms in (Liu et al., 2016), and developing the Stein geometry (Duncan et al., 2023; Nüsken & Renger, 2023) in the hybrid kernel setting. An open problem in the study of vanilla SVGD is finding a convergence rate in the finite particle regime. Other areas for future work are exploration of different kernel choices, a detailed study of scaling choices, and incorporation of the hybrid kernel setting with other SVGD variants such as matrix SVGD (Wang et al., 2019) or message passing SVGD (Zhuo et al., 2018; Wang et al., 2018).

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

## A    PROOFS

*Proof of Theorem 4.1.* Let $\phi \in \mathcal{B}$. For $1 \le i \le d$, we have the representations

$$\phi_i(\boldsymbol{x}) = \langle \phi_i, k_1(\boldsymbol{x}, \cdot) \rangle_{\mathcal{H}_1},$$
$$\phi_i(\boldsymbol{x}) = \langle \phi_i, k_2(\boldsymbol{x}, \cdot) \rangle_{\mathcal{H}_2}.$$

Theorem 3.1 from Liu & Wang (2016) applies to the transform $\boldsymbol{T}(\boldsymbol{x}) = \boldsymbol{x} + \epsilon\phi(\boldsymbol{x})$ for sufficiently small $\epsilon$ and sufficiently smooth $\phi$, so

$$
\begin{aligned}
-\nabla_\epsilon \mathrm{KL}(\boldsymbol{T}_{\#}\mu \parallel \nu_p)|_{\epsilon=0} &= \mathbb{E}_{\boldsymbol{x}\sim\mu}\left[ \boldsymbol{s}_p(\boldsymbol{x})^\top \phi(\boldsymbol{x}) + \nabla \cdot \phi(\boldsymbol{x}) \right] \\
&= \mathbb{E}_{\boldsymbol{x}\sim\mu}\left[ \boldsymbol{s}_p(\boldsymbol{x})^\top \left( \langle \phi_1, k_1(\boldsymbol{x}, \cdot) \rangle_{\mathcal{H}_1}, \ldots, \langle \phi_d, k_1(\boldsymbol{x}, \cdot) \rangle_{\mathcal{H}_1} \right) \right] \\
&\quad + \mathbb{E}_{\boldsymbol{x}\sim\mu}\left[ \nabla \cdot \left( \langle \phi_1, k_2(\boldsymbol{x}, \cdot) \rangle_{\mathcal{H}_2}, \ldots, \langle \phi_d, k_2(\boldsymbol{x}, \cdot) \rangle_{\mathcal{H}_2} \right) \right] \\
&= \mathbb{E}_{\boldsymbol{x}\sim\mu}\left[ \langle \phi, k_1(\boldsymbol{x}, \cdot)\boldsymbol{s}_p(\boldsymbol{x}) \rangle_{\mathcal{H}_1^d} \right] + \mathbb{E}_{\boldsymbol{x}\sim\mu}\left[ \langle \phi, \nabla_{\boldsymbol{x}} k_2(\boldsymbol{x}, \cdot) \rangle_{\mathcal{H}_2^d} \right] \\
&= \langle \phi, \boldsymbol{G}(\,\cdot\,; k_1, \mu, p) \rangle_{\mathcal{H}_1^d} + \langle \phi, \boldsymbol{R}(\,\cdot\,; k_2, \mu) \rangle_{\mathcal{H}_2^d} \\
&= \langle (\phi, \phi), (\boldsymbol{G}(\,\cdot\,; k_1, \mu, p), \boldsymbol{R}(\,\cdot\,; k_2, \mu)) \rangle_{\mathcal{H}_1^d \oplus \mathcal{H}_2^d} \\
&= \langle 2\phi, \phi_{\mu,p}^{k_1,k_2} \rangle_{\mathcal{H}_1^d + \mathcal{H}_2^d}.
\end{aligned}
$$

The above also uses the derivative reproducing properties (Zhou, 2008). Applying Lemma B.1 in the final step requires $\phi \neq 0$. But in the $\phi = 0$ case, $\mathrm{KL}(\boldsymbol{T}_{\#}\mu \parallel \nu_p) = \mathrm{KL}(\mu \parallel \nu_p)$ is constant with respect to $\epsilon$, so

$$-\nabla_\epsilon \mathrm{KL}(\boldsymbol{T}_{\#}\mu \parallel \nu_p)|_{\epsilon=0} = 0 = \langle 0, \phi_{\mu,p}^{k_1,k_2} \rangle_{\mathcal{H}_1^d + \mathcal{H}_2^d}.$$

Now by the Cauchy-Schwarz inequality,

$$-\nabla_{\epsilon}\mathrm{KL}(\boldsymbol{T}_{\#}\mu \parallel \mu)|_{\epsilon=0} \leq \left\|\phi_{\mu,p}^{k_1,k_2}\right\|_{\mathcal{H}_1^d + \mathcal{H}_2^d}$$

since $\phi \in \mathcal{B}$, with equality when $\phi = \phi_{\mu,p}^{k_1,k_2} / \left\|2\phi_{\mu,p}^{k_1,k_2}\right\|_{\mathcal{H}_1^d + \mathcal{H}_2^d}$. $\qquad\square$

**Remark 3.** *The above proof implicitly uses the fact that*

$$\mathbb{E}_{\boldsymbol{x}\sim\mu}\left[\langle\phi, f(\boldsymbol{x}, \cdot)\rangle_{\mathcal{H}_i}\right] = \langle\phi, \mathbb{E}_{\boldsymbol{x}\sim\mu}f(\boldsymbol{x}, \cdot)\rangle_{\mathcal{H}_i} \text{ for each } \phi \in \mathcal{H}_i$$

*for $i = 1, 2$. The interchange of expectation and inner product can be justified using Bochner integrals. For details, see (A.32) in (Steinwart & Christmann, 2008). To our knowledge, this is not made explicit in the literature despite being required in the proofs of results such as (Liu et al., 2016, Theorem 3.8) and (Liu, 2017, Theorem 3.5).*

For ease of notation in the following proof, we omit $\mu_\ell$ and $p$ when writing vector function components. In particular, for the update direction we write

$$\phi_{\mu_\ell,p}^{k_1,k_2}(\boldsymbol{x}) = (\phi_1^{k_1,k_2}(\boldsymbol{x}), \dots, \phi_d^{k_1,k_2}(\boldsymbol{x}))$$

with similar shorthands for the single kernel variants.

*Proof of Theorem 4.2.* We first prove the result for $\mathcal{H}_2 \subseteq \mathcal{H}_1$. Following the proof in Liu (2017),

$$\begin{aligned}
&\mathrm{KL}(\mu_{\ell+1} \parallel \nu_p) - \mathrm{KL}(\mu_\ell \parallel \nu_p) \\
&\leq -\epsilon_\ell \mathbb{E}_{\boldsymbol{x}\sim\mu_\ell}\left[\mathcal{S}_p\phi_{\mu_\ell,p}^{k_1,k_2}(\boldsymbol{x})\right] \\
&\quad + \epsilon_\ell^2 \mathbb{E}_{\boldsymbol{x}\sim\mu_\ell}\left[\frac{1}{2}\|\nabla\log p\|_{\mathrm{Lip}} \cdot \left\|\phi_{\mu_\ell,p}^{k_1,k_2}(\boldsymbol{x})\right\|_2^2 + 2\left\|\nabla\phi_{\mu_\ell,p}^{k_1,k_2}(\boldsymbol{x})\right\|_F^2\right] \\
&\leq -\epsilon_\ell \mathbb{E}_{\boldsymbol{x}\sim\mu_\ell}\left[\mathcal{S}_p\phi_{\mu_\ell,p}^{k_1}(\boldsymbol{x}) + \mathcal{S}_p R_1(\boldsymbol{x})\right] \\
&\quad + \epsilon_\ell^2 \mathbb{E}_{\boldsymbol{x}\sim\mu_\ell}\left[\frac{1}{2}\|\nabla\log p\|_{\mathrm{Lip}} \cdot \left\|\phi_{\mu_\ell,p}^{k_1,k_2}(\boldsymbol{x})\right\|_2^2 + 2\left\|\nabla\phi_{\mu_\ell,p}^{k_1,k_2}(\boldsymbol{x})\right\|_F^2\right] \\
&\leq -\epsilon_\ell \mathbb{S}_{k_1}(\mu_\ell, \nu_p) \\
&\quad + \epsilon_\ell^2 \mathbb{E}_{\boldsymbol{x}\sim\mu_\ell}\left[\frac{1}{2}\|\nabla\log p\|_{\mathrm{Lip}} \cdot \left\|\phi_{\mu_\ell,p}^{k_1,k_2}(\boldsymbol{x})\right\|_2^2 + 2\left\|\nabla\phi_{\mu_\ell,p}^{k_1,k_2}(\boldsymbol{x})\right\|_F^2 - \mathcal{S}_p R_1(\boldsymbol{x})\right]. \quad (14)
\end{aligned}$$

Note that the first inequality above relies the condition $\epsilon_\ell = (2\sup_x \rho(\nabla\phi^* + \nabla\phi^{*\top}))^{-1}$, which is in the statement of Theorem 4.2. For details, see (Liu, 2017, Appendix A.2).

We now bound $\left\|\phi_{\mu_\ell,p}^{k_1,k_2}(\boldsymbol{x})\right\|_2^2$ and $\left\|\nabla\phi_{\mu_\ell,p}^{k_1,k_2}(\boldsymbol{x})\right\|_F^2$. Assuming that $\mathcal{H}_2 \subseteq \mathcal{H}_1$, then $\phi_i^{k_1,k_2} \in \mathcal{H}_1^d$ for each $1 \leq i \leq n$ by Remark 1, and therefore

$$\begin{aligned}
\left\|\phi_i^{k_1,k_2}\right\|_{\mathcal{H}_1+\mathcal{H}_2}^2 &\leq \left\|\phi_i^{k_1,k_2}\right\|_{\mathcal{H}_1}^2 + \|0\|_{\mathcal{H}_2}^2 \\
&\leq \left(\left\|\phi_i^{k_1}\right\|_{\mathcal{H}_1} + \|\mathbb{E}_{\boldsymbol{y}\sim\mu_\ell}\left[\nabla_{y_i}\Delta k(\boldsymbol{y}, \cdot)\right]\|_{\mathcal{H}_1}\right)^2 \\
&\leq 2\left(\left\|\phi_i^{k_1}\right\|_{\mathcal{H}_1}^2 + \|\mathbb{E}_{\boldsymbol{y}\sim\mu_\ell}\left[\nabla_{y_i}\Delta k(\boldsymbol{y}, \cdot)\right]\|_{\mathcal{H}_1}^2\right) \quad (15)
\end{aligned}$$

by Equation (1), the triangle inequality, and the inequality $(a+b)^2 \leq 2(a^2+b^2)$, $a, b \in \mathbb{R}$.

Using the reproducing property, the Cauchy-Schwarz inequality, the definition of the algebraic sum RKHS norm (Berlinet & Thomas-Agnan, 2011, Theorem 5) (also see Equation (1)), and the bound

in (15), the first term in the remainder of (14) can be bounded by

$$
\begin{aligned}
\left\|\phi_{\mu_\ell,p}^{k_1,k_2}(\boldsymbol{x})\right\|_2^2 &= \sum_{i=1}^d \phi_i^{k_1,k_2}(\boldsymbol{x})^2 \\
&= \sum_{i=1}^d \left\langle (k_1+k_2)(\boldsymbol{x},\cdot), \phi_i^{k_1,k_2} \right\rangle_{\mathcal{H}_1+\mathcal{H}_2}^2 \\
&\le \sum_{i=1}^d \|(k_1+k_2)(\boldsymbol{x},\cdot)\|_{\mathcal{H}_1+\mathcal{H}_2}^2 \left\|\phi_i^{k_1,k_2}\right\|_{\mathcal{H}_1+\mathcal{H}_2}^2 \\
&\le 2(k_1+k_2)(\boldsymbol{x},\boldsymbol{x})\left(\mathbb{S}_{k_1}(\mu_\ell,\nu_p) + \|R_1(\boldsymbol{x})\|_{\mathcal{H}_1^d}^2\right).
\end{aligned}
\tag{16}
$$

Similarly, the second term in the remainder of (14) can be bounded by

$$
\begin{aligned}
\left\|\nabla\phi_{\mu_\ell,p}^{k_1,k_2}(\boldsymbol{x})\right\|_F^2 &= \sum_{i,j} \partial_{x_j}\phi_i^{k_1,k_2}(\boldsymbol{x})^2 \\
&= \sum_{i,j} \left\langle \partial_{x_j}(k_1+k_2)(\boldsymbol{x},\cdot), \phi_i^{k_1,k_2} \right\rangle_{\mathcal{H}_1+\mathcal{H}_2}^2 \\
&\le \sum_{i,j} \left\|\partial_{x_j}(k_1+k_2)(\boldsymbol{x},\cdot)\right\|_{\mathcal{H}_1+\mathcal{H}_2}^2 \left\|\phi_i^{k_1,k_2}\right\|_{\mathcal{H}_1+\mathcal{H}_2}^2 \\
&\le \sum_{j=1}^d \left(\partial_{x_j,x_j'}(k_1+k_2)(\boldsymbol{x},\boldsymbol{x}')|_{\boldsymbol{x}=\boldsymbol{x}'}\right)\sum_{i=1}^d \left\|\phi_i^{k_1,k_2}\right\|_{\mathcal{H}_1+\mathcal{H}_2}^2 \\
&\le 2\nabla_{\boldsymbol{x},\boldsymbol{x}'}(k_1+k_2)(\boldsymbol{x},\boldsymbol{x})\left(\mathbb{S}_{k_1}(\mu_\ell,\nu_p) + \|R_1(\boldsymbol{x})\|_{\mathcal{H}_1^d}^2\right)
\end{aligned}
\tag{17}
$$

where the derivative reproducing property (Zhou, 2008, Theorem 1 (b)) is also used.

Using $\nabla_{\boldsymbol{x},\boldsymbol{y}}$ as a shorthand for the operator $\sum_{i,j} \partial_{x_i}\partial_{y_j}$ and defining $\|\boldsymbol{f}\|_{\mu_\ell} := \mathbb{E}_{\boldsymbol{x}\sim\mu_\ell}|f_i(\boldsymbol{x})|$, we can bound the final terms in the remainder of (14) by

$$
\begin{aligned}
\mathbb{E}_{\boldsymbol{x}\sim\mu_\ell}\left[-\mathcal{S}_p R_1(\boldsymbol{x})\right] &\le \mathbb{E}_{\boldsymbol{x}\sim\mu_\ell}\left[\mathbb{E}_{\boldsymbol{y}\sim\mu_\ell}\left[\left|\boldsymbol{s}_p(\boldsymbol{x})^\top\nabla_{\boldsymbol{y}}\Delta k(\boldsymbol{y},\boldsymbol{x})\right| + |\nabla_{\boldsymbol{x},\boldsymbol{y}}\Delta k(\boldsymbol{y},\boldsymbol{x})|\right]\right] \\
&\le \|\boldsymbol{s}_p\|_{\mu_\ell} \sup_{\boldsymbol{x},\boldsymbol{y}} |\nabla_{\boldsymbol{y}}\Delta k(\boldsymbol{y},\boldsymbol{x})| + \sup_{\boldsymbol{x},\boldsymbol{y}}|\nabla_{\boldsymbol{x},\boldsymbol{y}}\Delta k(\boldsymbol{y},\boldsymbol{x})|.
\end{aligned}
\tag{18}
$$

The first result follows from combining all the above numbered equations where the constants are

$$
\begin{aligned}
C_1 &:= \sup_{\boldsymbol{x}}\left\{\|\nabla\log p\|_{\mathrm{Lip}}(k_1+k_2)(\boldsymbol{x},\boldsymbol{x}) + 4\nabla_{\boldsymbol{x},\boldsymbol{x}'}(k_1+k_2)(\boldsymbol{x},\boldsymbol{x})\right\}, \\
C_2 &:= \|\boldsymbol{s}_p\|_{\mu_\ell} \sup_{\boldsymbol{x},\boldsymbol{y}}|\nabla_{\boldsymbol{y}}\Delta k(\boldsymbol{y},\boldsymbol{x})| + \sup_{\boldsymbol{x},\boldsymbol{y}}|\nabla_{\boldsymbol{x},\boldsymbol{y}}\Delta k(\boldsymbol{y},\boldsymbol{x})|.
\end{aligned}
$$

When $\mathcal{H}_1 \subseteq \mathcal{H}_2$, we follow the same process as above with the decomposition $\phi_{\mu_\ell,p}^{k_1,k_2} = \phi_{\mu_\ell,p}^{k_2} + R_2$ instead of $\phi_{\mu_\ell,p}^{k_1,k_2} = \phi_{\mu_\ell,p}^{k_1} + R_1$. In this case, the constant $C_1$ is the same, but

$$
C_2 := \|\boldsymbol{s}_p\|_{\mu_\ell}^2 \sup_{\boldsymbol{x},\boldsymbol{y}}|\Delta k(\boldsymbol{y},\boldsymbol{x})| + \|\boldsymbol{s}_p\|_{\mu_\ell} \sup_{\boldsymbol{x},\boldsymbol{y}}|\nabla_{\boldsymbol{y}}\Delta k(\boldsymbol{y},\boldsymbol{x})|.
$$

$\square$

*Proof of Proposition 4.3.* The proof follows in the same way as that of (Gorham et al., 2020, Theorem 7) with the hybrid Stein operator in place of the usual Stein operator. In particular, the Wasserstein pseudo-Lipschitz property of h-SVGD can be established using the same technique as (Gorham et al., 2020, Lemma 12). $\square$

*Proof of Proposition 4.4.* This is a formal proof recalling the proof in (Liu, 2017, Appendix A.2). Take the statements of Theorem 4.2, divide by $\epsilon_\ell$, and take the limit as $\epsilon_\ell \to 0$. $\square$

*Proof of Proposition 4.5.* This proof follows as in (Liu, 2017, Theorem 3.5) with the reproducing properties applied in the algebraic sum RKHS. □

*Proof of Proposition 4.6.* The proof largely follows that of Theorem 2.4 Lu et al. (2019) with some minor adjustments. Notably, after fixing $r > 0$ and defining

$$Y_r := \left\{ u \in Y : \sup_{\boldsymbol{x} \in \mathcal{X}} |u(\boldsymbol{x}) - \boldsymbol{x}| < r \right\}$$

and the complete metric space

$$S_r := C([0, T_0]; Y_r),$$
$$d_S(u, v) := \sup_{t \in [0, T_0]} d_Y(u(t), v(t))$$

for some sufficiently small $T_0$ (to be determined later), the operator $\mathcal{F}$ must be modified to act on $u \in S_r$ via

$$\mathcal{F}(u)(t, \boldsymbol{x}) = \boldsymbol{x} - \int_0^t \int_{\mathcal{X}} \nabla K_2(u(s, \boldsymbol{x}) - u(s, \boldsymbol{x}'))\nu(d\boldsymbol{x}')ds$$
$$- \int_0^t \int_{\mathcal{X}} K_1(u(s, \boldsymbol{x}) - u(s, \boldsymbol{x}'))\nabla V(u(s, \boldsymbol{x}'))\nu(d\boldsymbol{x}')ds.$$

The relaxed Assumption (**B**1) and the same techniques of (Lu et al., 2019) are sufficient establish the required bounds to show that $\mathcal{F}$ is a contraction on $S_r$ for sufficiently small $T_0$. So the unique fixed point $X(\cdot, \cdot; \nu) \in S_r$ of $\mathcal{F}$ solves (13) in the interval $[0, T_0]$.

The $\min(\|\nabla K_1\|_\infty, \|\nabla K_2\|_\infty)$ term emerges because the telescoping in (Lu et al., 2019, Equation (3.8)) can be performed with either kernel. The remainder of the proof follows (Lu et al., 2019, Theorems 3.2 and 2.4). □

*Proof of Proposition 4.7.* This follows identically to the proof of (Korba et al., 2020, Proposition 7) with Lemma B.2 in place of Lemma 14. □

## B  Auxiliary Results

**Lemma B.1.** *For any $f_1, g_1 \in \mathcal{H}_1$ and $f_2, g_2 \in \mathcal{H}_2$ where $f_1 + f_2 \neq 0$,*

$$\langle (f_1, f_2), (g_1, g_2) \rangle_{\mathcal{H}_1 \oplus \mathcal{H}_2} = \langle f_1 + f_2, g_1 + g_2 \rangle_{\mathcal{H}_1 + \mathcal{H}_2}$$

*Proof.* Consider the map $u : \mathcal{H}_1 \oplus \mathcal{H}_2 \to \mathcal{H}_1 + \mathcal{H}_2$ given by $u(\phi_1, \phi_2) = \phi_1 + \phi_2$. Let its kernel be denoted by $N = u^{-1}(\{0\})$ and let $N^\perp$ be the orthogonal complement of $N$. The restriction of $u$ to $N^\perp$, denoted by $v$, is one-to-one and can be used to define an inner product via

$$\langle \phi, \psi \rangle_{\mathcal{H}_1 + \mathcal{H}_2} = \langle v^{-1}(\phi), v^{-1}(\psi) \rangle_{\mathcal{H}_1 \oplus \mathcal{H}_2},$$

making $\mathcal{H}_1 + \mathcal{H}_2$ an RKHS with kernel $k_1 + k_2$ (see (Berlinet & Thomas-Agnan, 2011, Theorem 5)). The assumption $f_1 + f_1 \neq 0$ implies that $(f_1, f_2) \in N^\perp$ and so

$$(f_1, f_2) = v^{-1}(f_1 + f_2).$$

The decomposition $\mathcal{H}_1 \oplus \mathcal{H}_2 = N + N^\perp$ gives

$$(g_1, g_2) = (g_1^N, g_2^N) + v^{-1}(g_1 + g_2).$$

Since $(g_1^N, g_2^N) \in N$ and $v^{-1}(f_1 + f_2) \in N^\perp$,

$$\langle (f_1, f_2), (g_1, g_2) \rangle_{\mathcal{H}_1 \oplus \mathcal{H}_2} = \langle v^{-1}(f_1 + f_2), (g_1^N, g_2^N) \rangle_{\mathcal{H}_1 \oplus \mathcal{H}_2} + \langle v^{-1}(f_1 + f_2), v^{-1}(g_1 + g_2) \rangle_{\mathcal{H}_1 \oplus \mathcal{H}_2}$$
$$= \langle f_1 + f_2, g_1 + g_2 \rangle_{\mathcal{H}_1 + \mathcal{H}_2}.$$

□

**Remark 4.** *The assumption that $f_1 + f_2 \neq 0$ is necessary. Otherwise the right hand side of the above will contain an inner product of two terms in $N$, retaining the terms $\langle f_1^N, g_1^N \rangle_{\mathcal{H}_1}$ and $\langle f_2^N, g_2^N \rangle_{\mathcal{H}_2}$.*

**Lemma B.2.** *Under assumptions (A1), (A4), (B1), (B2) the map*

$$(\boldsymbol{z}, \mu) \mapsto E(\boldsymbol{z}, \mu) := \int_{\mathcal{X}} -k_1(\boldsymbol{x}, \boldsymbol{z}) \nabla V(\boldsymbol{x}) + \nabla_{\boldsymbol{x}} k_2(\boldsymbol{x}, \boldsymbol{z}) d\mu(\boldsymbol{x})$$

*is $L$-Lipschitz. That is,*

$$\|E(\boldsymbol{z}, \mu) - E(\boldsymbol{z}', \mu')\|_2 \leq L(\|\boldsymbol{z} - \boldsymbol{z}'\|_2 + W_2(\mu, \mu'))$$

*where $L > 0$ depends on $k_1$, $k_2$ and $V$.*

*Proof.* Largely following the proof of Lemma 14 Korba et al. (2020), choosing an optimal coupling $s$ of $\mu$ and $\mu'$,

$$
\begin{aligned}
\|E(\boldsymbol{z}, \mu) - E(\boldsymbol{z}', \mu')\|_2 &\leq \big|\big| \mathbb{E}_s \left[ \nabla V(\boldsymbol{x}) (k_1(\boldsymbol{x}, \boldsymbol{z}) - k_1(\boldsymbol{x}', \boldsymbol{z}')) \right] \\
&\quad + \mathbb{E}_s \left[ (\nabla V(\boldsymbol{x}') - \nabla V(\boldsymbol{x})) k_1(\boldsymbol{x}', \boldsymbol{z}') \right] \\
&\quad + \mathbb{E}_s \left[ \nabla_{\boldsymbol{x}} k_2(\boldsymbol{x}, \boldsymbol{z}) - \nabla_{\boldsymbol{x}'} k_2(\boldsymbol{x}', \boldsymbol{z}) \right] \big|\big| \\
&\leq D \mathbb{E}_s \left[ \|\boldsymbol{x} - \boldsymbol{x}'\|_2 + \|\boldsymbol{z} - \boldsymbol{z}'\|_2 \right] \\
&\quad + BM \mathbb{E}_s \left[ \|\boldsymbol{x} - \boldsymbol{x}'\|_2 \right] \\
&\quad + D \mathbb{E}_s \left[ \|\boldsymbol{x} - \boldsymbol{x}'\|_2 + \|\boldsymbol{z} - \boldsymbol{z}'\|_2 \right] \\
&\leq (2D + BM) \left( \|\boldsymbol{z} - \boldsymbol{z}'\|_2 + W_2(\mu, \mu') \right).
\end{aligned}
$$

Note that the second term is bounded using the relaxed Assumption (**B2**) and there is no need to require that $|V|$ is bounded by a constant. $\square$

## C    DISCUSSION ON THE H-KSD

In this section, we discuss why we defined the h-KSD in its variational form instead of using one of the three other forms in (Liu et al., 2016). We also demonstrate that the h-KSD is not a valid discrepancy measure. We use the notation $\mathbb{S}_k(\mu, \nu_p)$ for the KSD with respect to $k$, and $\mathbb{S}_{k_1, k_2}(\mu, \nu_p)$ for the h-KSD with respect to $(k_1, k_2)$ defined in Equation (10).

Theorem 3.6 of (Liu et al., 2016) states that $\mathbb{S}_k(\mu, \nu_p) = \mathbb{E}_{\boldsymbol{x}, \boldsymbol{y} \sim \mu} \left[ \mathcal{S}_p^{\boldsymbol{x}} \mathcal{S}_p^{\boldsymbol{y}} \otimes k(\boldsymbol{x}, \boldsymbol{y}) \right]$ with $\boldsymbol{x}, \boldsymbol{y}$ independent. If this were to be taken as a starting point for defining a h-KSD, the natural generalisation would be $\mathbb{E}_{\boldsymbol{x}, \boldsymbol{y} \sim \mu} \left[ \mathcal{S}_p^{\boldsymbol{x}} \mathcal{S}_p^{\boldsymbol{y}} \otimes (k_1, k_2)(\boldsymbol{x}, \boldsymbol{y}) \right]$. However, this quantity can be negative for even simple choices of $k_1, k_2, \nu_p$ and $\mu$.

**Example 1.** *Let $\nu_p$ and $\mu$ be probability measures on $\mathcal{X} = \mathbb{R}$ with normal densities $\mathcal{N}(x; 0, 1)$ and $\mathcal{N}(x; 0, \sigma)$ respectively for $\sigma > 0$. Let $k_1$ and $k_2$ be RBF kernels with bandwidths $h_1$ and $h_2$ respectively. Define $\mathbb{S}_{k_1, k_2}^*(\mu, \nu_p) := \mathbb{E}_{\boldsymbol{x}, \boldsymbol{y} \sim \mu} \left[ \mathcal{S}_p^{\boldsymbol{x}} \mathcal{S}_p^{\boldsymbol{y}} \otimes (k_1, k_2)(\boldsymbol{x}, \boldsymbol{y}) \right]$ and note that $h_1 = h_2$ implies $\mathbb{S}_{k_1, k_2}^* = \mathbb{S}_{k_1} = \mathbb{S}_{k_2}$. Figure 3 shows a plot of this quantity for $\sigma \in [0.7, 1.3]$ for three combinations of $h_1$ and $h_2$. The $h_1 \neq h_2$ case demonstrates that $\mathbb{S}_{k_1, k_2}^*$ can be negative.*

The KSD also has as a spectral decomposition. Theorem 3.7 of (Liu et al., 2016) uses the fact that $\mathcal{S}_p^{\boldsymbol{x}} \mathcal{S}_p^{\boldsymbol{y}} \otimes (k_1, k_2)(\boldsymbol{x}, \boldsymbol{y})$ is a positive definite kernel as a function of $\boldsymbol{x}$ and $\boldsymbol{y}$. However, this quantity can be negative for some values of $\boldsymbol{x}$ and $\boldsymbol{y}$ since its expectation can be negative, as shown in the previous example. So it is not a positive definite kernel, and we cannot apply Mercer's theorem to attain the spectral decomposition.

We now turn to the original KSD definition of Equation (4) (Liu et al., 2016, Definition 3.2). Since the h-SVGD theory has been developed in the algebraic sum RKHS $\mathcal{H}_1 + \mathcal{H}_2$ whose kernel is $k_1 + k_2$, it may seem natural to propose

$$\mathbb{E}_{\boldsymbol{x}, \boldsymbol{y} \sim \mu} \left[ (\boldsymbol{s}_p(\boldsymbol{x}) - \boldsymbol{s}_q(\boldsymbol{x}))^\top (k_1 + k_2)(\boldsymbol{x}, \boldsymbol{y}) (\boldsymbol{s}_p(\boldsymbol{y}) - \boldsymbol{s}_q(\boldsymbol{y})) \right] \tag{19}$$

as a definition for the h-KSD. This of course simplifies to $\mathbb{S}_{k_1}(\mu, \nu_p) + \mathbb{S}_{k_2}(\mu, \nu_p)$. It may make sense to choose the average of the kernels in the above expression instead of their sum so that the

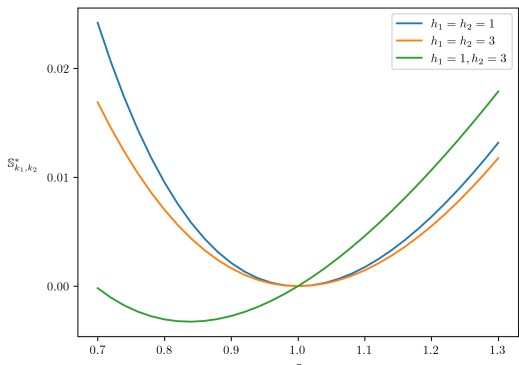

Figure 3: A simple example on $\mathcal{X} = \mathbb{R}$ showing the tractable form of the KSD (Liu & Wang, 2016, Theorem 3.6) does not extend to the hybrid kernel setting as a discrepancy measure, as it can take negative values.

h-KSD reduces to the KSD in the single kernel setting, that is, $\mathbb{S}_{k,k} = \mathbb{S}_k$. In fact the RKHS with kernels $k_1 + k_2$ and $\frac{1}{2}(k_1 + k_2)$ contain the same functions and have equivalent norms. However, from Theorem 4.1 and the subsequent discussion, we have

$$
\begin{aligned}
\mathbb{S}_{k_1,k_2}(\mu, \nu_p) &= \frac{\mathbb{E}_{\boldsymbol{x} \sim \mu}\left[\mathcal{S}_p \phi_{\mu,p}^{k_1,k_2}(\boldsymbol{x})\right]^2}{\left\|2\phi_{\mu,p}^{k_1,k_2}\right\|_{\mathcal{H}_1^2 + \mathcal{H}_2^d}^2} \\
&= \frac{\mathbb{E}_{\boldsymbol{x},\boldsymbol{y} \sim \mu}\left[\mathcal{S}_p^{\boldsymbol{x}} \mathcal{S}_p^{\boldsymbol{y}} \otimes (k_1, k_2)(\boldsymbol{x}, \boldsymbol{y})\right]^2}{\left\|2\phi_{\mu,p}^{k_1,k_2}\right\|_{\mathcal{H}_1^2 + \mathcal{H}_2^d}^2}
\end{aligned}
\tag{20}
$$

and it is not clear how this expression can be reconciled with (19).

We now address the issue of whether the h-KSD is a valid discrepancy measure. Recall that a kernel $k$ is integrally strictly positive definite if

$$
\int_{\mathcal{X}} \int_{\mathcal{X}} g(\boldsymbol{x}) k(\boldsymbol{x}, \boldsymbol{y}) g(\boldsymbol{y}) d\boldsymbol{x} d\boldsymbol{y} > 0
$$

for any function $g$ satisfying $0 < \|g\|_2^2 < \infty$ (Liu et al., 2016, Definition 3.1).

**Proposition C.1.** *For any measures $\mu$ and $\nu_p$ with continuous densities $q$ and $p$ respectively, we have $\mathbb{S}_{k_1,k_2}(\mu, \nu_p) \geq 0$. Suppose in addition that the kernels $k_1$ and $k_2$ are integrally strictly positive definite, $\|p(\cdot)(\boldsymbol{s}_p(\cdot) - \boldsymbol{s}_q(\cdot))\|_2^2 < \infty$, and either $\mathcal{H}_1 \subseteq \mathcal{H}_2$ or $\mathcal{H}_2 \subseteq \mathcal{H}_1$. Then $\mu = \nu_p$ implies $\mathbb{S}_{k_1,k_2}(\mu, \nu_p) = 0$.*

*Proof.* The variational definition in Equation (10) immediately ensures that $\mathbb{S}_{k_1,k_2}(\mu, \nu_p) \geq 0$.

We prove the second statement in the $\mathcal{H}_2 \subseteq \mathcal{H}_1$ case and note that the $\mathcal{H}_1 \subseteq \mathcal{H}_2$ case follows in the same manner. Suppose that $\mu = \nu_p$ almost everywhere. Note that $\phi_{\mu,p}^{k_1,k_2} \in \mathcal{H}_1$ since $\mathcal{H}_2 \subseteq \mathcal{H}_1$. Using this along with the first part of this result, Equation (1), and (Liu et al., 2016, Proposition 3.3), we have

$$
\begin{aligned}
0 &\leq \mathbb{S}_{k_1,k_2}(\mu, \nu_p) \\
&= \left\|\phi_{\mu,p}^{k_1,k_2}\right\|_{\mathcal{H}_1^d + \mathcal{H}_2^d}^2 \\
&\leq \left\|\phi_{\mu,p}^{k_1,k_2}\right\|_{\mathcal{H}_1^d}^2 + \|0\|_{\mathcal{H}_2^d}^2 \\
&= \mathbb{S}_{k_1}(\mu, \nu_p) \\
&= 0.
\end{aligned}
$$

$\square$

We caution that this result is not an extension of (Liu et al., 2016, Proposition 3.3) to the hybrid kernel setting. In particular, the converse of the second statement is false.

**Proposition C.2.** *Under the assumptions of Proposition C.1,* $\mathbb{S}_{k_1,k_2}(\mu,\nu_p) = 0$ *does not imply* $\mu = \nu_p$ *almost everywhere.*

*Proof.* Example 1 showed that $\mathbb{E}_{\boldsymbol{x},\boldsymbol{y}\sim\mu}\left[\mathcal{S}_p^{\boldsymbol{x}}\mathcal{S}_p^{\boldsymbol{y}}\otimes(k_1,k_2)(\boldsymbol{x},\boldsymbol{y})\right]$ may be zero even when $\mu$ and $\nu_p$ are not equal almost everywhere. This is sufficient for a counterexample in light of (20). ☐

Although $\mathbb{S}_{k_1,k_2}$ is not a valid discrepancy measure, this does not cause problems for the algorithm because Proposition 4.2 and Remark 2 ensure a rate of decrease in the KL divergence of $\mathbb{S}_{k_1}$ or $\mathbb{S}_{k_2}$ for standard choices of $k_1$ and $k_2$.

We leave a deeper investigation of the h-KSD for future research. Interesting directions would be reconciling our variational definition with the original definition in (Liu et al., 2016) and drawing connections to the Stein geometry (Liu, 2017; Nüsken & Renger, 2023; Duncan et al., 2023).

## D EXPERIMENTS

This section contains further details on the numerical experiments presented in Section 5 and additional numerical results.

Recall that the RBF kernel is defined as

$$k_{\mathrm{RBF}}(\boldsymbol{x},\boldsymbol{y};h) := \exp\left(-\frac{\|\boldsymbol{x}-\boldsymbol{y}\|_2^2}{2h}\right), \qquad h > 0$$

where $h$ is the bandwidth. It is common practice throughout the SVGD literature (Liu & Wang, 2016) to set the bandwidth to $h_{\mathrm{med}} := \mathrm{med}^2/\log(n)$ where $\mathrm{med}$ is the median pairwise distance between particles $(\boldsymbol{x}_i)_{i=1}^n$. In Section 5, we set $k_1 = k_{\mathrm{RBF}}(\cdot,\cdot;h_{\mathrm{med}})$ with a stronger repulsive kernel $k_2 = f(d)k_1$, choosing $f(d)$ to be either $\sqrt{d}$ or $\log(d)$.

In this appendix, we extend experiments in Sections 5.1 and 5.2 with additional results where the bandwidth of the RBF kernel scales with the dimension. In particular, with $h_1$ and $h_2$ the bandwidths of $k_1$ and $k_2$ respectively, we set $h_1 = h_{\mathrm{med}}$ and $h_2 = f(d)h_{\mathrm{med}}$. Intuitively, a larger bandwidth on the repulsive kernel will ensure a slower decay of $\nabla_{\boldsymbol{x}}k_{\mathrm{RBF}}(\boldsymbol{x},\boldsymbol{y})$ as $\|\boldsymbol{x}-\boldsymbol{y}\|_2$ increases. This would increase the magnitude of the repulsive force, $\|\boldsymbol{R}(\,\cdot\,;k_2,\mu)\|_\infty$, thereby enabling more distant particles to still repel each other.

We also provide additional results for the BNN experiment in 5.2 with other forms of the repulsive kernel. These include the inverse multi-quadratic (IMQ) kernel $k_{\mathrm{IMQ}}$ (Gorham & Mackey, 2017)

$$k_{\mathrm{IMQ}}(\boldsymbol{x},\boldsymbol{y};h,c,\beta) := \left(c^2 + \frac{\|\boldsymbol{x}-\boldsymbol{y}\|_2^2}{2h}\right)^\beta \qquad h > 0, c > 0, \beta < 0,$$

the Laplace kernel $k_{\mathrm{Lap}}$ (Sriperumbudur et al., 2010)

$$k_{\mathrm{Lap}}(\boldsymbol{x},\boldsymbol{y};h) := \exp\left(-\frac{\|\boldsymbol{x}-\boldsymbol{y}\|_2}{h}\right) \qquad h > 0,$$

and the inverse log kernel $k_{\mathrm{IL}}$ (Chen et al., 2018)

$$k_{\mathrm{IL}}(\boldsymbol{x},\boldsymbol{y};h) := \left(h^{-2} + \ln\left(1 + \|\boldsymbol{x}-\boldsymbol{y}\|_2^2\right)\right)^{-1} \qquad h > 0.$$

In experiments using the IMQ kernel, we set $c = 1$ and $\beta \in \{-0.5, -1\}$.

Following D'Angelo et al. (2021), we also define a functional RBF kernel $k_{\mathrm{RBF(f)}}$ as follows. Let $P$ denote the number of features, let $M$ denote the number of records in the training dataset, let $F \leq M$ denote the number of records to be used in the functional kernel evaluation, and recall that $d$ is the number of weights in the BNN. Let $B : \mathcal{X}^P \times \mathcal{X}^d \to \mathbb{R}$ denote the neural network where

we use $B_{\boldsymbol{x}}(\boldsymbol{u}) = B(\boldsymbol{u}, \boldsymbol{x})$ to denote the evaluation of the network with weights $\boldsymbol{x}$ on the data point $\boldsymbol{u}$. From the training dataset, choose a subset of data points $\boldsymbol{u}_1, \ldots, \boldsymbol{u}_F \in \mathcal{X}^P$ and define

$$k_{\mathrm{RBF(f)}}(\boldsymbol{x}, \boldsymbol{y}; h) := \frac{1}{F} \sum_{i=1}^{F} k_{\mathrm{RBF}}\left(B_{\boldsymbol{x}}(\boldsymbol{u}_i), B_{\boldsymbol{y}}(\boldsymbol{u}_i); h\right).$$

The intuition behind choosing this as a repulsive kernel for the BNN experiment is that is should repel particles that correspond to network weights that generate similar outputs. Our implementation uses $F = \min(30, M)$ and randomly selects $F$ records from the training dataset at each iteration. The bandwidth is chosen using the median heuristic, but with the network outputs instead. That is, let $\mathrm{med(f)}$ denote the median pairwise distance between $(B_{\boldsymbol{x}_i}(\boldsymbol{u}_1), \ldots, B_{\boldsymbol{x}_i}(\boldsymbol{u}_F))_{i=1}^{n}$, where this distance is in $\mathbb{R}^F$, and define $h_{\mathrm{med(f)}} := \mathrm{med(f)}^2 / \log(n)$.

Since $k_{\mathrm{RBF}}$ decays exponentially with respect to the weights and $k_{\mathrm{RBF(f)}}$ decays exponentially with respect to the network outputs, one must decay faster than the other, at least outside some compact set of the weight space $\mathcal{X}^d$. Therefore, Remark 2 ensures that either $\mathcal{H}_1 \subseteq \mathcal{H}_2$ or $\mathcal{H}_2 \subseteq \mathcal{H}_1$ will hold, and the conditions of Theorem 4.2 are satisfied.

### D.1 Variance Collapse in the Proportional Limit

The results of this experiment in Section 5.2 used 1000 iterations of h-SVGD on a sequence of multivariate normal (MVN) distributions with dimensions ranging from $d = 5, \ldots, d = 200$. Let $p_d, \boldsymbol{\mu}_d$ and $\boldsymbol{\Sigma}_d$ denote the density, mean vector, and covariance matrix of the $d$-dimensional MVN. We choose the means and covariances such that the marginal moments in lower dimensions are equal across each $p_d$. That is, $(\boldsymbol{\mu}_d)_i = (\boldsymbol{\mu}_{d'})_i$ and $(\boldsymbol{\Sigma}_d)_{ij} = (\boldsymbol{\Sigma}_{d'})_{ij}$ for all $1 \leq i, j \leq d < d'$. Following (Ba et al., 2021), the number of particles $N$ is chosen such that $d/N$ approaches a proportional limit $\gamma$. We present results in three schemes, $\gamma < 1$, $\gamma = 1$ and $\gamma > 1$. Each configuration is averaged over 10 runs, each with a different set of initial particles independently drawn from a standard multivariate normal distribution.

Comparing Figure 4 to Figure 1 shows that bandwidth scaling is not as effective as weight scaling at mitigating variance collapse. Logarithmic bandwidth scaling in the $\gamma < 1$ scheme is the exception, with noticeable overestimation of the variance. In all cases, bandwidth scaling provided poorer mean estimates than vanilla SVGD, suggesting weight scaling to be a more suitable choice in the proportional limit.

### D.2 Bayesian Neural Network

The results presented in Section 5.2 follow the settings of (Liu & Wang, 2016). In particular, we use normal priors for the network weights and a Gamma prior for the inverse covariances. There is one hidden layer with 50 units for most datasets, Protein and Year being the exceptions with 100 units each. The datasets are randomly partitioned into $90\%$ for training and $10\%$ for testing with results averaged over 20 trials, Protein and Year being the exceptions with 5 trials and 1 trial respectively. The number of particles in each case is 20, the activation function is $\mathrm{RELU}(x) = \max(0, x)$, the number of iterations is 2000, and the mini-batch size is 100 for all datasets except for Year, which uses a mini-batch size of 1000.

The repulsive kernels in Section 5.2 were chosen to be $k_2 = \log(d) \cdot k_1$ and $k_2 = \sqrt{d} \cdot k_1$ with $k_1 = k_{\mathrm{RBF}}(\cdot, \cdot; h_{\mathrm{med}})$. This demonstrated a consistent improvement in the DAMV and comparable results on the test RMSE and LL metrics.

On the other hand, choosing bandwidths $h_1 = h_{\mathrm{med}}$ and $h_2 = \sqrt{d} \cdot h_{\mathrm{med}}$ can lead to minor improvements in test LL scores on most datasets when compared to vanilla SVGD at the expense of no real improvement in the DAMV.

We include results for repulsive kernels from families other than RBF, namely $k_{\mathrm{IMQ}}$, $k_{\mathrm{Lap}}$ and $k_{\mathrm{IL}}$, to demonstrate that the h-SVGD update will still lead to comparable results. Results with a functional RBF kernel, $k_{\mathrm{RBF(f)}}$ have also been included.

Figure 5 summarises the results of the experiments described above, and the values are reported in Tables 1-10. This figure demonstrates that for a variety of repulsive kernels, h-SVGD yields RMSE

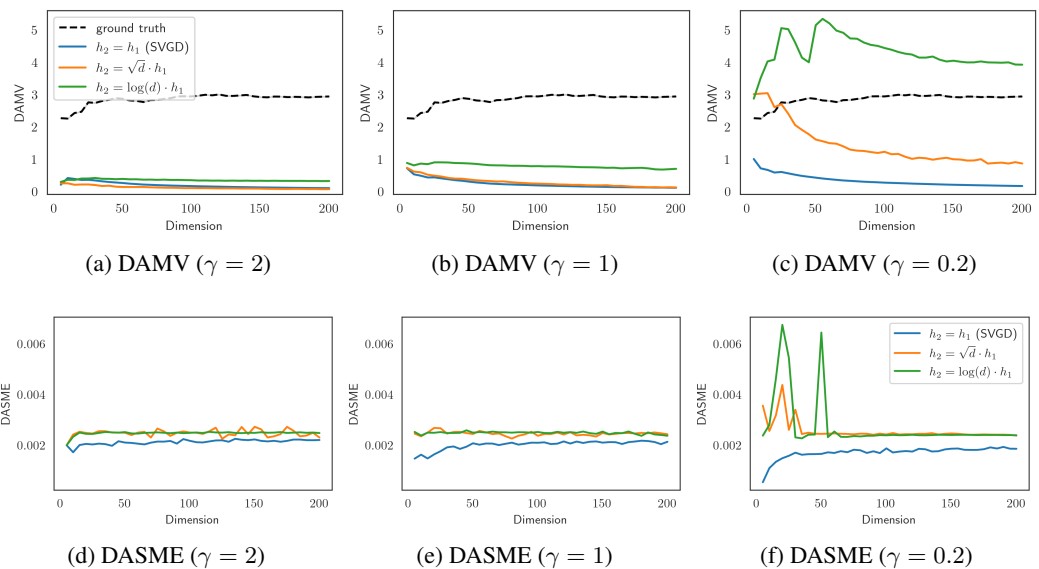

(a) DAMV ($\gamma = 2$)    (b) DAMV ($\gamma = 1$)    (c) DAMV ($\gamma = 0.2$)

(d) DASME ($\gamma = 2$)    (e) DASME ($\gamma = 1$)    (f) DASME ($\gamma = 0.2$)

Figure 4: DAMV and DASME of MVN distributions in the proportional limit with bandwidth scaling.

and LL metrics comparable to vanilla SVGD, and that certain repulsive kernels can improve the DAMV in high dimensions.

Table 1: Average DAMV and test performance (RMSE and LL) with standard errors on the Boston dataset with $k_1 = k_{\mathrm{RBF}}(\cdot, \cdot; h_{\mathrm{med}})$.

| $k_2$ | DAMV | Test RMSE | Test LL |
|---|---|---|---|
| $k_{\mathrm{RBF}}(\cdot, \cdot; h_{\mathrm{med}})$ (SVGD) | $0.067 \pm 0.003$ | $2.968 \pm 0.051$ | $-2.656 \pm 0.019$ |
| $\log(d) \cdot k_{\mathrm{RBF}}(\cdot, \cdot; h_{\mathrm{med}})$ | $0.098 \pm 0.008$ | $3.015 \pm 0.069$ | $-2.669 \pm 0.022$ |
| $\sqrt{d} \cdot k_{\mathrm{RBF}}(\cdot, \cdot; h_{\mathrm{med}})$ | $0.247 \pm 0.021$ | $3.160 \pm 0.092$ | $-2.722 \pm 0.034$ |
| $k_{\mathrm{RBF}}(\cdot, \cdot; \log(d) \cdot h_{\mathrm{med}})$ | $0.073 \pm 0.003$ | $2.992 \pm 0.055$ | $-2.667 \pm 0.023$ |
| $k_{\mathrm{RBF}}(\cdot, \cdot; \sqrt{d} \cdot h_{\mathrm{med}})$ | $0.064 \pm 0.003$ | $2.984 \pm 0.071$ | $-2.658 \pm 0.030$ |
| $k_{\mathrm{IMQ}}(\cdot, \cdot; h_{\mathrm{med}}, 1, 0.5)$ | $0.059 \pm 0.003$ | $2.958 \pm 0.045$ | $-2.651 \pm 0.018$ |
| $k_{\mathrm{IMQ}}(\cdot, \cdot; h_{\mathrm{med}}, 1, 1)$ | $0.058 \pm 0.003$ | $2.979 \pm 0.060$ | $-2.660 \pm 0.026$ |
| $k_{\mathrm{IL}}(\cdot, \cdot; h_{\mathrm{med}})$ | $0.062 \pm 0.005$ | $2.959 \pm 0.055$ | $-2.651 \pm 0.017$ |
| $k_{\mathrm{Lap}}(\cdot, \cdot; h_{\mathrm{med}})$ | $0.064 \pm 0.003$ | $2.984 \pm 0.071$ | $-2.658 \pm 0.031$ |
| $k_{\mathrm{RBF(f)}}(\cdot, \cdot; h_{\mathrm{med(f)}})$ | $0.060 \pm 0.004$ | $2.959 \pm 0.031$ | $-2.650 \pm 0.018$ |

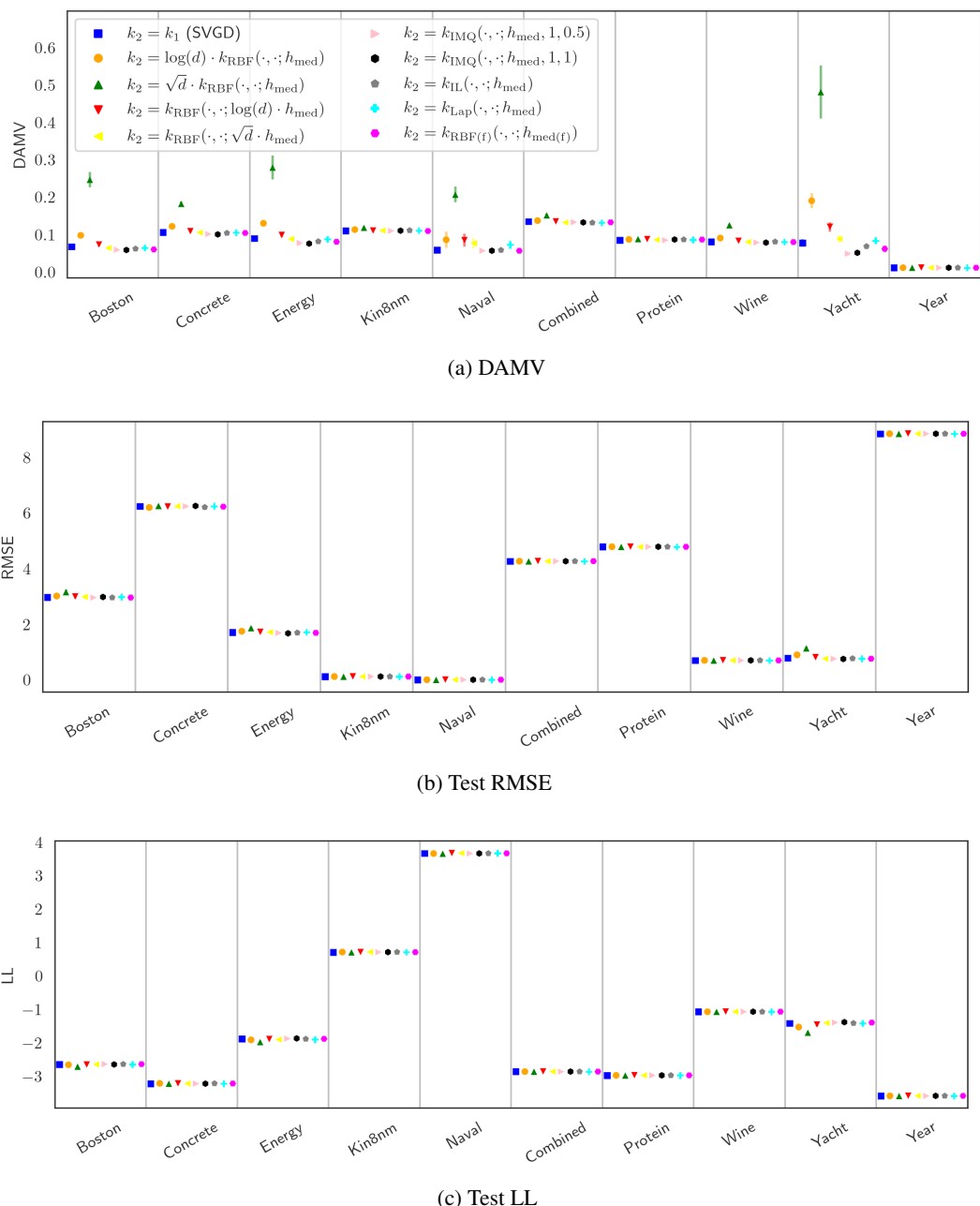

(a) DAMV

(b) Test RMSE

(c) Test LL

Figure 5: DAMV, RMSE and LL metrics with standard errors for several different repulsive kernels.

Table 2: Average DAMV and test performance (RMSE and LL) with standard errors on the Concrete dataset with $k_1 = k_{\mathrm{RBF}}(\cdot, \cdot; h_{\mathrm{med}})$.

| $k_2$ | DAMV | Test RMSE | Test LL |
|---|---|---|---|
| $k_{\mathrm{RBF}}(\cdot, \cdot; h_{\mathrm{med}})$ (SVGD) | $0.106 \pm 0.002$ | $6.237 \pm 0.056$ | $-3.232 \pm 0.011$ |
| $\log(d) \cdot k_{\mathrm{RBF}}(\cdot, \cdot; h_{\mathrm{med}})$ | $0.122 \pm 0.004$ | $6.197 \pm 0.067$ | $-3.225 \pm 0.012$ |
| $\sqrt{d} \cdot k_{\mathrm{RBF}}(\cdot, \cdot; h_{\mathrm{med}})$ | $0.182 \pm 0.006$ | $6.257 \pm 0.046$ | $-3.232 \pm 0.008$ |
| $k_{\mathrm{RBF}}(\cdot, \cdot; \log(d) \cdot h_{\mathrm{med}})$ | $0.108 \pm 0.003$ | $6.224 \pm 0.060$ | $-3.230 \pm 0.010$ |
| $k_{\mathrm{RBF}}(\cdot, \cdot; \sqrt{d} \cdot h_{\mathrm{med}})$ | $0.105 \pm 0.004$ | $6.241 \pm 0.054$ | $-3.231 \pm 0.009$ |
| $k_{\mathrm{IMQ}}(\cdot, \cdot; h_{\mathrm{med}}, 1, 0.5)$ | $0.101 \pm 0.002$ | $6.239 \pm 0.055$ | $-3.232 \pm 0.011$ |
| $k_{\mathrm{IMQ}}(\cdot, \cdot; h_{\mathrm{med}}, 1, 1)$ | $0.100 \pm 0.003$ | $6.251 \pm 0.044$ | $-3.234 \pm 0.008$ |
| $k_{\mathrm{IL}}(\cdot, \cdot; h_{\mathrm{med}})$ | $0.104 \pm 0.003$ | $6.206 \pm 0.066$ | $-3.227 \pm 0.012$ |
| $k_{\mathrm{Lap}}(\cdot, \cdot; h_{\mathrm{med}})$ | $0.105 \pm 0.004$ | $6.242 \pm 0.054$ | $-3.232 \pm 0.009$ |
| $k_{\mathrm{RBF(f)}}(\cdot, \cdot; h_{\mathrm{med(f)}})$ | $0.104 \pm 0.004$ | $6.225 \pm 0.052$ | $-3.229 \pm 0.009$ |

Table 3: Average DAMV and test performance (RMSE and LL) with standard errors on the Energy dataset with $k_1 = k_{\mathrm{RBF}}(\cdot, \cdot; h_{\mathrm{med}})$.

| $k_2$ | DAMV | Test RMSE | Test LL |
|---|---|---|---|
| $k_{\mathrm{RBF}}(\cdot, \cdot; h_{\mathrm{med}})$ (SVGD) | $0.090 \pm 0.006$ | $1.706 \pm 0.043$ | $-1.896 \pm 0.034$ |
| $\log(d) \cdot k_{\mathrm{RBF}}(\cdot, \cdot; h_{\mathrm{med}})$ | $0.130 \pm 0.009$ | $1.749 \pm 0.051$ | $-1.924 \pm 0.042$ |
| $\sqrt{d} \cdot k_{\mathrm{RBF}}(\cdot, \cdot; h_{\mathrm{med}})$ | $0.279 \pm 0.032$ | $1.866 \pm 0.041$ | $-1.986 \pm 0.030$ |
| $k_{\mathrm{RBF}}(\cdot, \cdot; \log(d) \cdot h_{\mathrm{med}})$ | $0.098 \pm 0.005$ | $1.718 \pm 0.045$ | $-1.904 \pm 0.039$ |
| $k_{\mathrm{RBF}}(\cdot, \cdot; \sqrt{d} \cdot h_{\mathrm{med}})$ | $0.088 \pm 0.004$ | $1.718 \pm 0.038$ | $-1.914 \pm 0.039$ |
| $k_{\mathrm{IMQ}}(\cdot, \cdot; h_{\mathrm{med}}, 1, 0.5)$ | $0.077 \pm 0.005$ | $1.690 \pm 0.042$ | $-1.887 \pm 0.036$ |
| $k_{\mathrm{IMQ}}(\cdot, \cdot; h_{\mathrm{med}}, 1, 1)$ | $0.076 \pm 0.005$ | $1.674 \pm 0.033$ | $-1.881 \pm 0.036$ |
| $k_{\mathrm{IL}}(\cdot, \cdot; h_{\mathrm{med}})$ | $0.081 \pm 0.005$ | $1.696 \pm 0.054$ | $-1.897 \pm 0.043$ |
| $k_{\mathrm{Lap}}(\cdot, \cdot; h_{\mathrm{med}})$ | $0.088 \pm 0.004$ | $1.719 \pm 0.039$ | $-1.915 \pm 0.040$ |
| $k_{\mathrm{RBF(f)}}(\cdot, \cdot; h_{\mathrm{med(f)}})$ | $0.081 \pm 0.005$ | $1.691 \pm 0.032$ | $-1.894 \pm 0.030$ |

Table 4: Average DAMV and test performance (RMSE and LL) with standard errors on the Kin8nm dataset with $k_1 = k_{\mathrm{RBF}}(\cdot, \cdot; h_{\mathrm{med}})$.

| $k_2$ | DAMV | Test RMSE | Test LL |
|---|---|---|---|
| $k_{\mathrm{RBF}}(\cdot, \cdot; h_{\mathrm{med}})$ (SVGD) | $0.110 \pm 0.002$ | $0.120 \pm 0.001$ | $0.699 \pm 0.011$ |
| $\log(d) \cdot k_{\mathrm{RBF}}(\cdot, \cdot; h_{\mathrm{med}})$ | $0.113 \pm 0.002$ | $0.120 \pm 0.001$ | $0.702 \pm 0.007$ |
| $\sqrt{d} \cdot k_{\mathrm{RBF}}(\cdot, \cdot; h_{\mathrm{med}})$ | $0.118 \pm 0.002$ | $0.121 \pm 0.001$ | $0.698 \pm 0.010$ |
| $k_{\mathrm{RBF}}(\cdot, \cdot; \log(d) \cdot h_{\mathrm{med}})$ | $0.110 \pm 0.002$ | $0.121 \pm 0.001$ | $0.697 \pm 0.010$ |
| $k_{\mathrm{RBF}}(\cdot, \cdot; \sqrt{d} \cdot h_{\mathrm{med}})$ | $0.110 \pm 0.003$ | $0.120 \pm 0.001$ | $0.703 \pm 0.010$ |
| $k_{\mathrm{IMQ}}(\cdot, \cdot; h_{\mathrm{med}}, 1, 0.5)$ | $0.110 \pm 0.002$ | $0.120 \pm 0.001$ | $0.699 \pm 0.011$ |
| $k_{\mathrm{IMQ}}(\cdot, \cdot; h_{\mathrm{med}}, 1, 1)$ | $0.110 \pm 0.002$ | $0.120 \pm 0.001$ | $0.700 \pm 0.010$ |
| $k_{\mathrm{IL}}(\cdot, \cdot; h_{\mathrm{med}})$ | $0.111 \pm 0.002$ | $0.120 \pm 0.001$ | $0.702 \pm 0.007$ |
| $k_{\mathrm{Lap}}(\cdot, \cdot; h_{\mathrm{med}})$ | $0.110 \pm 0.003$ | $0.120 \pm 0.001$ | $0.703 \pm 0.010$ |
| $k_{\mathrm{RBF(f)}}(\cdot, \cdot; h_{\mathrm{med(f)}})$ | $0.109 \pm 0.002$ | $0.121 \pm 0.001$ | $0.698 \pm 0.007$ |

Table 5: Average DAMV and test performance (RMSE and LL) with standard errors on the Naval dataset with $k_1 = k_{\mathrm{RBF}}(\cdot, \cdot; h_{\mathrm{med}})$.

| $k_2$ | DAMV | Test RMSE | Test LL |
|---|---|---|---|
| $k_{\mathrm{RBF}}(\cdot, \cdot; h_{\mathrm{med}})$ (SVGD) | $0.059 \pm 0.004$ | $0.006 \pm 0.000$ | $3.654 \pm 0.006$ |
| $\log(d) \cdot k_{\mathrm{RBF}}(\cdot, \cdot; h_{\mathrm{med}})$ | $0.086 \pm 0.021$ | $0.006 \pm 0.000$ | $3.648 \pm 0.006$ |
| $\sqrt{d} \cdot k_{\mathrm{RBF}}(\cdot, \cdot; h_{\mathrm{med}})$ | $0.207 \pm 0.021$ | $0.006 \pm 0.000$ | $3.648 \pm 0.005$ |
| $k_{\mathrm{RBF}}(\cdot, \cdot; \log(d) \cdot h_{\mathrm{med}})$ | $0.084 \pm 0.017$ | $0.006 \pm 0.000$ | $3.657 \pm 0.006$ |
| $k_{\mathrm{RBF}}(\cdot, \cdot; \sqrt{d} \cdot h_{\mathrm{med}})$ | $0.076 \pm 0.013$ | $0.006 \pm 0.000$ | $3.660 \pm 0.008$ |
| $k_{\mathrm{IMQ}}(\cdot, \cdot; h_{\mathrm{med}}, 1, 0.5)$ | $0.057 \pm 0.001$ | $0.006 \pm 0.000$ | $3.651 \pm 0.006$ |
| $k_{\mathrm{IMQ}}(\cdot, \cdot; h_{\mathrm{med}}, 1, 1)$ | $0.057 \pm 0.001$ | $0.006 \pm 0.000$ | $3.651 \pm 0.005$ |
| $k_{\mathrm{IL}}(\cdot, \cdot; h_{\mathrm{med}})$ | $0.058 \pm 0.004$ | $0.006 \pm 0.000$ | $3.653 \pm 0.005$ |
| $k_{\mathrm{Lap}}(\cdot, \cdot; h_{\mathrm{med}})$ | $0.072 \pm 0.012$ | $0.006 \pm 0.000$ | $3.659 \pm 0.008$ |
| $k_{\mathrm{RBF(f)}}(\cdot, \cdot; h_{\mathrm{med(f)}})$ | $0.056 \pm 0.001$ | $0.006 \pm 0.000$ | $3.654 \pm 0.007$ |

Table 6: Average DAMV and test performance (RMSE and LL) with standard errors on the Combined dataset with $k_1 = k_{\mathrm{RBF}}(\cdot, \cdot; h_{\mathrm{med}})$.

| $k_2$ | DAMV | Test RMSE | Test LL |
|---|---|---|---|
| $k_{\mathrm{RBF}}(\cdot, \cdot; h_{\mathrm{med}})$ (SVGD) | $0.135 \pm 0.003$ | $4.266 \pm 0.004$ | $-2.873 \pm 0.001$ |
| $\log(d) \cdot k_{\mathrm{RBF}}(\cdot, \cdot; h_{\mathrm{med}})$ | $0.137 \pm 0.002$ | $4.266 \pm 0.004$ | $-2.873 \pm 0.001$ |
| $\sqrt{d} \cdot k_{\mathrm{RBF}}(\cdot, \cdot; h_{\mathrm{med}})$ | $0.152 \pm 0.004$ | $4.264 \pm 0.003$ | $-2.872 \pm 0.001$ |
| $k_{\mathrm{RBF}}(\cdot, \cdot; \log(d) \cdot h_{\mathrm{med}})$ | $0.135 \pm 0.003$ | $4.265 \pm 0.005$ | $-2.873 \pm 0.002$ |
| $k_{\mathrm{RBF}}(\cdot, \cdot; \sqrt{d} \cdot h_{\mathrm{med}})$ | $0.132 \pm 0.004$ | $4.266 \pm 0.005$ | $-2.873 \pm 0.001$ |
| $k_{\mathrm{IMQ}}(\cdot, \cdot; h_{\mathrm{med}}, 1, 0.5)$ | $0.133 \pm 0.003$ | $4.266 \pm 0.004$ | $-2.873 \pm 0.001$ |
| $k_{\mathrm{IMQ}}(\cdot, \cdot; h_{\mathrm{med}}, 1, 1)$ | $0.132 \pm 0.004$ | $4.265 \pm 0.003$ | $-2.873 \pm 0.001$ |
| $k_{\mathrm{IL}}(\cdot, \cdot; h_{\mathrm{med}})$ | $0.132 \pm 0.002$ | $4.266 \pm 0.004$ | $-2.873 \pm 0.001$ |
| $k_{\mathrm{Lap}}(\cdot, \cdot; h_{\mathrm{med}})$ | $0.132 \pm 0.004$ | $4.266 \pm 0.005$ | $-2.873 \pm 0.001$ |
| $k_{\mathrm{RBF(f)}}(\cdot, \cdot; h_{\mathrm{med(f)}})$ | $0.133 \pm 0.003$ | $4.269 \pm 0.004$ | $-2.874 \pm 0.001$ |

Table 7: Average DAMV and test performance (RMSE and LL) with standard errors on the Protein dataset with $k_1 = k_{\mathrm{RBF}}(\cdot, \cdot; h_{\mathrm{med}})$.

| $k_2$ | DAMV | Test RMSE | Test LL |
|---|---|---|---|
| $k_{\mathrm{RBF}}(\cdot, \cdot; h_{\mathrm{med}})$ (SVGD) | $0.085 \pm 0.001$ | $4.784 \pm 0.004$ | $-2.986 \pm 0.001$ |
| $\log(d) \cdot k_{\mathrm{RBF}}(\cdot, \cdot; h_{\mathrm{med}})$ | $0.087 \pm 0.001$ | $4.785 \pm 0.004$ | $-2.986 \pm 0.001$ |
| $\sqrt{d} \cdot k_{\mathrm{RBF}}(\cdot, \cdot; h_{\mathrm{med}})$ | $0.088 \pm 0.001$ | $4.785 \pm 0.006$ | $-2.986 \pm 0.001$ |
| $k_{\mathrm{RBF}}(\cdot, \cdot; \log(d) \cdot h_{\mathrm{med}})$ | $0.087 \pm 0.001$ | $4.783 \pm 0.003$ | $-2.986 \pm 0.001$ |
| $k_{\mathrm{RBF}}(\cdot, \cdot; \sqrt{d} \cdot h_{\mathrm{med}})$ | $0.086 \pm 0.001$ | $4.781 \pm 0.002$ | $-2.985 \pm 0.001$ |
| $k_{\mathrm{IMQ}}(\cdot, \cdot; h_{\mathrm{med}}, 1, 0.5)$ | $0.085 \pm 0.001$ | $4.784 \pm 0.004$ | $-2.986 \pm 0.001$ |
| $k_{\mathrm{IMQ}}(\cdot, \cdot; h_{\mathrm{med}}, 1, 1)$ | $0.087 \pm 0.001$ | $4.785 \pm 0.006$ | $-2.986 \pm 0.001$ |
| $k_{\mathrm{IL}}(\cdot, \cdot; h_{\mathrm{med}})$ | $0.086 \pm 0.001$ | $4.785 \pm 0.004$ | $-2.986 \pm 0.001$ |
| $k_{\mathrm{Lap}}(\cdot, \cdot; h_{\mathrm{med}})$ | $0.086 \pm 0.001$ | $4.781 \pm 0.002$ | $-2.985 \pm 0.001$ |
| $k_{\mathrm{RBF(f)}}(\cdot, \cdot; h_{\mathrm{med(f)}})$ | $0.086 \pm 0.001$ | $4.782 \pm 0.004$ | $-2.985 \pm 0.001$ |

Table 8: Average DAMV and test performance (RMSE and LL) with standard errors on the Wine dataset with $k_1 = k_{\mathrm{RBF}}(\cdot, \cdot; h_{\mathrm{med}})$.

| $k_2$ | DAMV | Test RMSE | Test LL |
|---|---|---|---|
| $k_{\mathrm{RBF}}(\cdot, \cdot; h_{\mathrm{med}})$ (SVGD) | $0.081 \pm 0.002$ | $0.701 \pm 0.002$ | $-1.082 \pm 0.004$ |
| $\log(d) \cdot k_{\mathrm{RBF}}(\cdot, \cdot; h_{\mathrm{med}})$ | $0.091 \pm 0.002$ | $0.702 \pm 0.002$ | $-1.084 \pm 0.004$ |
| $\sqrt{d} \cdot k_{\mathrm{RBF}}(\cdot, \cdot; h_{\mathrm{med}})$ | $0.125 \pm 0.003$ | $0.702 \pm 0.001$ | $-1.082 \pm 0.003$ |
| $k_{\mathrm{RBF}}(\cdot, \cdot; \log(d) \cdot h_{\mathrm{med}})$ | $0.083 \pm 0.002$ | $0.702 \pm 0.002$ | $-1.083 \pm 0.003$ |
| $k_{\mathrm{RBF}}(\cdot, \cdot; \sqrt{d} \cdot h_{\mathrm{med}})$ | $0.080 \pm 0.002$ | $0.701 \pm 0.002$ | $-1.082 \pm 0.004$ |
| $k_{\mathrm{IMQ}}(\cdot, \cdot; h_{\mathrm{med}}, 1, 0.5)$ | $0.078 \pm 0.002$ | $0.701 \pm 0.002$ | $-1.082 \pm 0.004$ |
| $k_{\mathrm{IMQ}}(\cdot, \cdot; h_{\mathrm{med}}, 1, 1)$ | $0.078 \pm 0.002$ | $0.701 \pm 0.002$ | $-1.082 \pm 0.004$ |
| $k_{\mathrm{IL}}(\cdot, \cdot; h_{\mathrm{med}})$ | $0.081 \pm 0.002$ | $0.702 \pm 0.002$ | $-1.084 \pm 0.003$ |
| $k_{\mathrm{Lap}}(\cdot, \cdot; h_{\mathrm{med}})$ | $0.080 \pm 0.002$ | $0.701 \pm 0.002$ | $-1.081 \pm 0.004$ |
| $k_{\mathrm{RBF(f)}}(\cdot, \cdot; h_{\mathrm{med(f)}})$ | $0.080 \pm 0.002$ | $0.701 \pm 0.002$ | $-1.081 \pm 0.004$ |

Table 9: Average DAMV and test performance (RMSE and LL) with standard errors on the Yacht dataset with $k_1 = k_{\mathrm{RBF}}(\cdot, \cdot; h_{\mathrm{med}})$.

| $k_2$ | DAMV | Test RMSE | Test LL |
|---|---|---|---|
| $k_{\mathrm{RBF}}(\cdot, \cdot; h_{\mathrm{med}})$ (SVGD) | $0.077 \pm 0.010$ | $0.785 \pm 0.040$ | $-1.430 \pm 0.036$ |
| $\log(d) \cdot k_{\mathrm{RBF}}(\cdot, \cdot; h_{\mathrm{med}})$ | $0.190 \pm 0.020$ | $0.899 \pm 0.042$ | $-1.541 \pm 0.043$ |
| $\sqrt{d} \cdot k_{\mathrm{RBF}}(\cdot, \cdot; h_{\mathrm{med}})$ | $0.481 \pm 0.071$ | $1.140 \pm 0.064$ | $-1.706 \pm 0.048$ |
| $k_{\mathrm{RBF}}(\cdot, \cdot; \log(d) \cdot h_{\mathrm{med}})$ | $0.120 \pm 0.012$ | $0.812 \pm 0.039$ | $-1.471 \pm 0.046$ |
| $k_{\mathrm{RBF}}(\cdot, \cdot; \sqrt{d} \cdot h_{\mathrm{med}})$ | $0.088 \pm 0.009$ | $0.761 \pm 0.030$ | $-1.418 \pm 0.035$ |
| $k_{\mathrm{IMQ}}(\cdot, \cdot; h_{\mathrm{med}}, 1, 0.5)$ | $0.049 \pm 0.005$ | $0.758 \pm 0.042$ | $-1.405 \pm 0.033$ |
| $k_{\mathrm{IMQ}}(\cdot, \cdot; h_{\mathrm{med}}, 1, 1)$ | $0.051 \pm 0.005$ | $0.752 \pm 0.047$ | $-1.394 \pm 0.034$ |
| $k_{\mathrm{IL}}(\cdot, \cdot; h_{\mathrm{med}})$ | $0.068 \pm 0.007$ | $0.771 \pm 0.026$ | $-1.425 \pm 0.031$ |
| $k_{\mathrm{Lap}}(\cdot, \cdot; h_{\mathrm{med}})$ | $0.083 \pm 0.008$ | $0.765 \pm 0.026$ | $-1.429 \pm 0.033$ |
| $k_{\mathrm{RBF(f)}}(\cdot, \cdot; h_{\mathrm{med(f)}})$ | $0.062 \pm 0.009$ | $0.761 \pm 0.062$ | $-1.407 \pm 0.052$ |

Table 10: Average DAMV and test performance (RMSE and LL) on the Year dataset with $k_1 = k_{\mathrm{RBF}}(\cdot, \cdot; h_{\mathrm{med}})$. No standard errors are included because there was only one trial run once for each kernel.

| $k_2$ | DAMV | Test RMSE | Test LL |
|---|---|---|---|
| $k_{\mathrm{RBF}}(\cdot, \cdot; h_{\mathrm{med}})$ (SVGD) | $0.011 \pm$ NA | $8.847 \pm$ NA | $-3.599 \pm$ NA |
| $\log(d) \cdot k_{\mathrm{RBF}}(\cdot, \cdot; h_{\mathrm{med}})$ | $0.011 \pm$ NA | $8.846 \pm$ NA | $-3.599 \pm$ NA |
| $\sqrt{d} \cdot k_{\mathrm{RBF}}(\cdot, \cdot; h_{\mathrm{med}})$ | $0.011 \pm$ NA | $8.847 \pm$ NA | $-3.599 \pm$ NA |
| $k_{\mathrm{RBF}}(\cdot, \cdot; \log(d) \cdot h_{\mathrm{med}})$ | $0.011 \pm$ NA | $8.847 \pm$ NA | $-3.599 \pm$ NA |
| $k_{\mathrm{RBF}}(\cdot, \cdot; \sqrt{d} \cdot h_{\mathrm{med}})$ | $0.011 \pm$ NA | $8.847 \pm$ NA | $-3.599 \pm$ NA |
| $k_{\mathrm{IMQ}}(\cdot, \cdot; h_{\mathrm{med}}, 1, 0.5)$ | $0.011 \pm$ NA | $8.847 \pm$ NA | $-3.599 \pm$ NA |
| $k_{\mathrm{IMQ}}(\cdot, \cdot; h_{\mathrm{med}}, 1, 1)$ | $0.011 \pm$ NA | $8.847 \pm$ NA | $-3.599 \pm$ NA |
| $k_{\mathrm{IL}}(\cdot, \cdot; h_{\mathrm{med}})$ | $0.011 \pm$ NA | $8.846 \pm$ NA | $-3.599 \pm$ NA |
| $k_{\mathrm{Lap}}(\cdot, \cdot; h_{\mathrm{med}})$ | $0.011 \pm$ NA | $8.846 \pm$ NA | $-3.599 \pm$ NA |
| $k_{\mathrm{RBF(f)}}(\cdot, \cdot; h_{\mathrm{med(f)}})$ | $0.011 \pm$ NA | $8.846 \pm$ NA | $-3.599 \pm$ NA |

