# OpenReview forum: "Hybrid Kernel Stein Variational Gradient Descent"
_ICLR.cc/2024/Conference — Submitted to ICLR 2024_

### Official Review · Reviewer_9Q7g · 2023-10-27

**Soundness:** 3 good
**Presentation:** 3 good
**Contribution:** 3 good
**Rating:** 6
**Confidence:** 3

**Summary:**

This paper analyzes the theoretical properties of hybrid Stein variational gradient descent, which uses a different kernel for the attraction term and the repulsion term. It is shown that h-SVGD update direction is optimal in decreasing the KL divergence. An assortment of results for h-SVGD are proved, extending recent theoretical results regarding SVGD. Experiments are performed to verify that h-SVGD can mitigate the variance collapse problem in high dimensions.

**Strengths:**

* The writing is very clear and generally good, except it seems unmotivated at times.
* The proofs are concise (I have only checked Thm 4.1 closely) and it is more careful (e.g. Remark 3) and has better generalization (e.g. Lemma B.2 and Proposition 4.7) compared to existing results.
* While the idea of analyzing the update direction in the sum of two RKHSes is natural, it is nevertheless clean and well-explained.
* The discussion in Appendix C on why h-KSD does not have an easily computable form (compared to KSD) is an important and interesting addition to the paper.
* Experiments seem sufficient in illustrating the advantage of h-SVGD in mitigating variance collapse.

**Weaknesses:**

* The analysis seems like straightforward extensions of existing results, e.g., most proofs in Section 4 are 1-liners.
* While the result (Thm 4.1) on the optimality of the h-SVGD update direction and many theoretical results does not require $k_1$ to be related to $k_2$, in all experiments, $k_2$ is simply a scalar multiple of $k_1$, i.e., $k_2 = f(d)k_1$. This seems to suggest the more general theory does not give rise to diverse choices of $k_1$ and $k_2$ in applications.
* Even if we only consider the case of $k_2 = f(d)k_1$, it remains a question of how to choose the scaling function $f(d)$. The paper suggests taking $f(d) = \sqrt{d}$ or $\ln(d)$, but further comparision (either empircal or theoretical) is lacking.
* As discussed in Appendix C, h-KSD is not a valid discrepancy measure, which seems to suggest it is less useful as a metric than the vanilla KSD.

**Questions:**

1. Could the authors explain why $\phi_{\mu, p}^{k_1,k_2} \in \mathcal{H}_1^d\cap \mathcal{H}_2^d$ in Thm 4.1?
2. Are there any applications of choices of $k_1$ and $k_2$ such that $\mathcal{H}_1$ and $\mathcal{H}_2$ does not include one another?
3. How is the bandwidth of the kernels affecting the variance collapse, compared to the choice of $f(d)$? Or to put the question in another way, how to simultaneously choose the bandwidth and $f(d)$ in applications?
4. At the beginning of Section 5, the authors mention [Zhuo et al. 2018] that puts a "conditional dependence structure". What does this mean exactly?
5. In Table 1, the test accuracy on Yacht for h-SVGD is poor compared to SVGD. Why is this the case?
6. Where is the map $\Phi_p^{k_1,k_2}$ defined in (9)?

---

> ### Author Response · Authors · 2023-11-20
>
> We thank the reviewer for their time in providing their review and feedback.
>
> **Weaknesses:**
>
> 1. We acknowledge that some proofs are presented as straightforward extensions and one-liners. However, we would like to mention some key technical details in the Appendix. Lemma B.1 is non-trivial and is required to establish the optimality of the h-SVGD update direction. Some oversights in proofs in the existing literature have been addressed. For example, our Assumption (B2) replaces a previous very restrictive assumption which, through Lemma B.2, makes Proposition 4.7 a stronger result than its counterpart in Korba et al. (2020). Also, Remark 3 addresses the interchangeability of expectation and inner products, which has not been addressed in the SVGD literature to our knowledge.
>
> 2. Diversity of kernel choice: As noted, the developed theory does not require kernels $k_1$ and $k_2$ to be related, but the simulation results were based on kernels of the form $k_2=f(d)k_1$. We have now included additional BNN experiments with a range different kernels in Appendix D. Please see Main Response for further details.
>
> 3. The current work serves to establish the theory of h-SVGD for the first time, and to demonstrate its ability to alleviate variance collapse. However, we agree that choosing the scaling function is a worthwhile direction for future research, and we now explicitly mention this in Section 5.2 and the conclusion.
>
> 4. While, as correctly noted, h-KSD is not a valid discrepancy measure, the descent lemma (Theorem 4.2) shows that the KL divergence bounds can be written in terms of the vanilla KSD and guarantees a decrease in the KL divergence at every step. Some additional discussion has been added to Section 4.3. See the Main Response for further discussion.
>
> **Questions:**
>
> 1. The set within which the optimisation takes place is chosen as a subset of this intersection. For standard kernel choices, Remark 2 ensures that this intersection will be either $\mathcal H_1$ or $\mathcal H_2$. This amounts to optimising within a unit ball on $\mathcal H_1$ or $\mathcal H_2$, although with a norm induced by the sum Hilbert space.
>
> 2. To our knowledge, the SVGD literature does not contain kernel choices of this type.
>
> 3. If $k_2$ is the RBF kernel, a larger bandwidth allows a steeper gradient $\nabla k_2$ when evaluated on distant particles. For intuition, think of the plot of $\exp(-x^2/h)$, so that a higher bandwidth increases the repulsive force, and thereby mitigates variance collapse. We now provide a comment on this in Appendix D.
>
> 4. The conditional dependence structure refers to probabilistic graphical models that factorise into a set of lower dimensional inference problems. In this example, variance collapse is alleviated by working in lower dimensions. We have slightly modified the beginning of Section 5 to make this more clear.
>
> 5. One possible reason is that the Yacht dataset has the smallest number of records (306) and the second smallest number of features (6) of the ten datasets considered.
>
> 6. The paper has been updated with the definition of $\Phi_p^{k_1,k_2}$ now just before equation (9).

---

> > ### Comment · Reviewer_9Q7g · 2023-11-21
> > **Reply to authors**
> >
> > Thank you for your clarification. I appreciate the effort in the paper revision and the additional BNN experiments. I think the idea of using h-SVGD to prevent variance collapse, while not new, is thoroughly invested in this work. At the same time, I'm not fully convinced that using h-SVGD is always better than SVGD. I think one main disadvantage is that there are more hyperparameters (choices of two kernels, kernel parameters, and scaling parameter $f(d)$), and better heuristics on how to choose these parameters are missing. As such, I would like to keep my current score.

---

### Official Review · Reviewer_pUod · 2023-10-30

**Soundness:** 3 good
**Presentation:** 2 fair
**Contribution:** 2 fair
**Rating:** 6
**Confidence:** 3

**Summary:**

The paper presents a theoretical justification for using h-SVGD, a variant of the Stein variational gradient descent (SVGD) method in which different kernels are used for the gradient term and the repulsive terms. The authors show that this method can mitigate the variance collapse problem without extra computational cost while remaining competitive to standard SVGD.

**Strengths:**

* The background section surveys the relevant studies and concepts for this paper well.
* The theoretical results in this paper seem to be novel and may be relevant for the community.
* The authors indeed demonstrate that the variance collapse phenomena is reduced to some extent according to the proposed metric.

**Weaknesses:**

* The paper focuses on h-SVGD, which is fine, but I am not convinced about the impact of this SVGD variant. The empirical results in this paper do not show a conclusive advantage for preferring this method over the standard SVGD, and the same applies to the original paper by D’Angelo et al., (2021).
* Following the last point, although the scope of the paper is to provide a theoretical ground for h-SVGD, perhaps it will have a stronger contribution if the authors would clearly state (and evaluate) families of valid kernels for the repulsive term.
* I find it odd that the test log-likelihood is not correlative with the dimension averaged marginal variance. If indeed the particles are more diverse with h-SVGD then I expected that it will be reflected in a better test log-likelihood.
* The method section is not written clearly enough in my opinion. Specifically, the authors can provide better intuition for some of the results. Also, perhaps the authors should present only the main claims in the main text and provide a proof sketch for them.

**Questions:**

* In D’Angelo et al., (2021) the authors used a different kernel for the repulsive term from the ones used in this paper. Is there something in the theory that does not apply on their kernel? It may be interesting to evaluate the performance and variance shrinkage of that kernel as well.

---

> ### Author Response · Authors · 2023-11-20
>
> We thank the reviewer for their time in providing their review and feedback.
>
> **Weaknesses:**
>
> 1. Significance of h-SVGD: Please see our Main Response for full details. We have now improved our presentation of our empirical results in Section 5 to more clearly demonstrate that h-SVGD has similar performance to SVGD in terms of test RMSE and test log-likelihood, but improved (or equivalent) variance estimation. This is now most clearly seen in Figure 1, revised Figure 2, and new Figure 5 in Appendix D, which summarises performance over 9 kernel variants.
>
> 2. Evaluation of kernel families:
> Thank you for this welcome suggestion. We have now added results using additional different kernel families (9 variants in total) for the repulsive kernel to Appendix D, including the kernel adopted in D'Angelo et al. (2021). The theory still applies to this latter kernel choice; in particular, the conditions for the descent lemma (Theorem 4.2) are satisfied, as detailed in Appendix D.
>
> 3. Log likelihood not correlated with dimension averaged marginal variance:
> To our understanding, log-likelihood is a measure of test accuracy. Better variance representation doesn't necessarily mean that log-likelihood is improved.
>
> 4. Better intuition:
> We have now included some additional commentary in Section 4 to provide more intuition on the results. In particular, we note Remark 1 and Remark 2 (which has now been moved slightly earlier in the paper) for some intuition on the main result (Theorem 4.1). Further discussion has also been added around Theorem 4.2 on how the KL divergence will always decrease under h-SVGD.
>
> **Questions:**
>
> 1. D'Angelo et al. (2021)'s kernel: As requested we have now included simulations using this kernel for the repulsive term, as outlined above.

---

> > ### Comment · Reviewer_pUod · 2023-11-22
> > **Response to Authors**
> >
> > I would like to thank the authors for their answers. I believe the authors addressed most of the concerns raised by me and other reviewers. I am still not convinced about the impact of this SVGD variant. Nevertheless, I acknowledge the value of this paper in terms of its theoretical contribution to the field and that it may help future studies, either theoretic or algorithmic ones. Therefore, I decided to raise my score to 6.

---

### Official Review · Reviewer_WGpG · 2023-10-31

**Soundness:** 3 good
**Presentation:** 2 fair
**Contribution:** 2 fair
**Rating:** 5
**Confidence:** 3

**Summary:**

This paper proposed a theoretical framework for Stein variational gradient descent with hybrid kernels in drift and repulsive terms. This paper mainly leverages the tools from the previous work to analyse the meaning of descent direction in SVGD, large time asymptotics, large particle limits and its gradient flow form. Empirically, the author conduct one synthetic and one Bayesian neural network.

**Strengths:**

The paper presents a theoretical framework for hybrid kernel SVGD. By leveraging the tools from previous work, the analysis is extensive. If the reader is familiar with the Stein discrepancy, the presentation is clear. Originality is not the strongest selling point of this paper, since the theoretical analysis follows from the previous work and extend the previous analysis to the hybrid kernel space, but it is still good to see the hybrid kernel trick has a proper underlying theory associated with it.

**Weaknesses:**

My primary concern pertains to the apparent significance of the hybrid kernel approach, as presented in the paper. The paper suggests that the hybrid kernel is proposed as a solution to circumvent the issue of variance collapsing. Nonetheless, it should be noted that there are numerous preceding studies such as S-SVGD, Grassman SVGD, among others, addressing similar challenges. Some of these methods have successfully established a proper goodness-of-fit test, ensuring that the resultant discrepancy is a valid one.
Despite this, I observed a lack of empirical evidence showcasing the hybrid kernel approach’s advantages over these established methods. In light of this, could you please elucidate on the specific benefits and improvements of the hybrid kernel approach, be it from a theoretical or empirical standpoint?

My second concern revolves around the convergence properties of the h-SVGD algorithm. The manuscript demonstrates that the descent magnitude is h-KSD, which, as acknowledged, is not a proper discrepancy. This raises questions regarding the algorithm’s capability to minimize the KL divergence effectively, specifically, whether it can drive the KL divergence to zero. A descent magnitude (h-KSD) of zero does not implies that the distributions are equal or that the KL divergence has been minimized to zero.
This brings us back to the previous point on the need for the hybrid kernel approach’s advantages. It is good to understand how h-SVGD, with its unique convergence characteristics, stands out amidst other existing methodologies addressing similar issues.

**Questions:**

1. For theorem 4.1, how do you ensure the $H_1 \cap H_2$ is not empty?
2. From the experiment 5.1, it seems that the variance still collapses but at a slower speed. But from the plot in S-SVGD or GSVGD paper, the variance estimation does not drop at $d=100$. So what is the advantages of the hybrid approach?

---

> ### Author Response · Authors · 2023-11-20
>
> We thank the reviewer for their time in providing their review and feedback.
>
> **Weaknesses:**
>
> 1. Significance of h-SVGD: Please see our Main Response for full details. We have now improved our presentation of our empirical results in Section 5 to more clearly demonstrate that h-SVGD has similar performance to SVGD in terms of test RMSE and test log-likelihood, but improved (or equivalent) variance estimation. This is now most clearly seen in Figure 1, revised Figure 2, and new Figure 5 in Appendix D, which summarises performance over 9 kernel variants. In addition, at the start of Section 5 we now comment that other previous methods (such as GSVGD and S-SVGD) also tackle variance collapse, but they do so at a far greater computational cost. In particular, S-SVGD requires additional computation of the optimal test directions and GSVGD requires additional computation at each step to update the projectors. The advantage of h-SVGD is that there is no added computational cost over regular SVGD.
>
> 2. Convergence properties of h-SVGD:
> The question is about whether h-SVGD can minimise the KL divergence effectively: see our Main Response for full details. As noted, the descent magnitude is h-KSD (Theorem 4.1), which is not a proper discrepancy. However, the descent lemma (Theorem 4.2) bounds the decrease in KL divergence in terms of a proper discrepancy, namely the KSD of one of the kernels. We have now included further details in Section 4.3 that show for proper choice of step size, the KL divergence is strictly decreasing at all times, meaning that the h-SVGD algorithm avoids cases where $\mathbb S_{k_1,k_2}(\mu_\ell^\infty,\nu_p)=0$ but $\mu_\ell^\infty$ and $\nu_p$ are not equal almost everywhere.
>
> **Questions:**
>
> 1. Remark 2 mentions that either $\mathcal H_1 \subseteq \mathcal H_2$ or $\mathcal H_2 \subseteq \mathcal H_1$ for many common choices of kernel (e.g. RBF, IMQ, log-inverse, or Matérn). This includes when $k_1$ and $k_2$ are from different families mentioned above. So the intersection $\mathcal H_1 \cap \mathcal H_2$ will be either $\mathcal H_1$ or $\mathcal H_2$. We are not aware of applications in the SVGD literature that use kernels other than those listed above. Remark 2 has been reworded slightly and moved so that it appears just after Theorem 4.1.
>
> 2. As mentioned in response to the weaknesses above, the advantage of h-SVGD is that it alleviates the variance collapse at no additional computational cost, whereas S-SVGD and GSVGD require additional computations of the optimal test direction or the projectors at each step.

---

> > ### Comment · Reviewer_WGpG · 2023-11-20
> >
> > **Additional computation cost of GSVGD and S-SVGD**: Yes, those methods require additional costs. However, if those method can obtain better performance or variance estimation, sometimes it is affordable to have this additional cost. That is why I want to see the performance comparison. At for drawing samples from simple Gaussian distribution, it seems that GSVGD and S-SGVD have more stable variance estimation?
> >
> > **Convergence properties**: I am still a bit confused here. The descent magnitude is not a proper discrepancy, right? So if the magnitude is 0, that means the distribution is no longer moving, but it does not mean it is equal to the target distribution. But from the descent lemma, you claim to recover the target distribution. So where is the discrepancy, do I misunderstand something?

---

### Official Review · Reviewer_5E3e · 2023-10-31

**Soundness:** 4 excellent
**Presentation:** 4 excellent
**Contribution:** 3 good
**Rating:** 6
**Confidence:** 4

**Summary:**

A hybrid kernel variant of SVGD is theoretically analysed in this paper. By defining a hybrid Stein operator and, subsequently, h-KSD, they prove that (1) the h-SVGD update direction is optimal within an appropriate RKHS, (2) h-SVGD guarantees a decrease in the KL divergence at each step and (3) other limit results. Experimentally, h-SVGD also mitigates the crucial variance collapse of SVGD algorithms at no additional cost and is shown to be competitive with other SVGD methods.

**Strengths:**

- h-SVGD has previously been proposed heuristically by D'Angelo et al. (2021). This paper provides a theoretical analysis of h-SVGD, which is a significant contribution to the literature: both the optimal update direction and the KL divergence decrease are important theoretical results for any new SVGD algorithm.
- The large time asymptotics of h-SVGD are analysed, showing that h-SVGD always decreases the KL and converges to the true posterior in the limit.
- Seemingly technical theoretical results are given adequate intuition and explanation, making the paper accessible to a wide audience, including applied users of SVGD algorithms.
- Most SVGD algorithms suffer from variance collapse, which is a significant issue in practice. Some results show h-SVGD is shown to mitigate this issue, which would be a significant practical contribution.

**Weaknesses:**

- Despite rigorous theoretical results, the experimental results are not sufficient to show that it mitigates the variance collapse issue better than previous methods (e.g. S-SVGD and G-SVGD). For (2), it would be useful to study the variance collapse issue with inference tasks in higher dimensions in comparison to previous approaches, such as the experiments in [1], as this is mainly an issue that arises in large dimensions.

**Questions:**

- What is the computational cost of h-SVGD compared to SVGD? Is it the same or more expensive?

---

> ### Author Response · Authors · 2023-11-20
>
> We thank the reviewer for their time in providing their review and feedback.
>
> **Weaknesses:**
>
> 1. Strong experimental results for h-SVGD: Please see our Main Response for full details. However, note that the mitigation of variance collapse in high dimensions is demonstrated in Section 5.1 and Figure 1 (a-c), which shows a significant improvement in estimating the true variance in high dimensions over SVGD, even when the number of dimensions is greater than the number of particles ($\gamma>1$). Also, revised Figure 2 now more clearly demonstrates an increase in DAMV for the BNN example, with consistent performance in test RMS and log-likelihood. Please also see new Figure 5 in Appendix D for additional qualitatively similar results with other kernel choices (9 kernel variations). Overall, the general performance of h-SVGD is comparable with SVGD but with the advantage that h-SVGD has superior performance in mitigating variance collapse.
>
> 2. While the primary aim of this paper is to provide theoretical justification of h-SVGD, we recognise that it is useful to support this by also demonstrating the performance of h-SVGD in areas that are directly related to this theoretical development; namely performance with regard to different families of kernels. This has now been done, and has been implemented in Appendix D. We feel that comparison to other competitor methods (such as GSVGD and S-SVGD) is somewhat adjacent to the focus of this paper, and is more relevant for research that makes algorithmic contributions in this field.
>
> **Questions:**
>
> 1. In terms of computational costs: the cost of updating the particles for both SVGD and h-SVGD is $O(N^2)$ at each step. We have added this statement at the beginning of Section 5. The SVGD update can be slightly faster when $k(x,y)$ can be reused to compute $\nabla k(x,y)$ (this applies to the RBF kernel for example), but the order is still $O(N^2)$.

---

### Author Response · Authors · 2023-11-20
**Main Response**

We thank each of the Reviewers for their time reviewing our paper, and their considered feedback. As a direct result, the following major changes have been made to the manuscript. (Note that smaller changes have also been made in response to each Reviewer's individual comments, as detailed in the individualised responses.)

* **The advantages of h-SVGD:** All reviewers note that the simulation results could better demonstrate the usefulness of h-SVGD over SVGD. While the primary contribution of this paper is to develop the theoretical underpinnings of h-SVGD, we agree that it would be useful to demonstrate the practical competitiveness of h-SVGD in a range of scenarios, particularly with a range of kernels. Here we note that the primary motivation and use-case for h-SVGD is in mitigating variance collapse. For this reason, we would expect exemplar h-SVGD results to demonstrate improved or equivalent estimation of the true model variance when compared to SVGD, with equivalent performance for other sampler aspects, such as test RMSE and test log-likelihood. To this end, we have made the following changes to the simulation results:

    - Figure 2 in Section 5.2 has been updated to include test RMSE and test log-likelihood (LL), with the original Table 1 being moved to Appendix D, as it was easier to understand the results in Figure form. It is now clear from the graphical representation that variance collapse (as measured by DAMV) is typically improved or mitigated with h-SVGD, with equivalent performances in test RMS and test log-likelihood. Section 5.1 and Figure 1 already presented strong evidence for h-SVGD performance in high dimensions (orange \& green lines) over SVGD (blue lines), even when the number of dimensions is greater than the number of particles ($\gamma>1$).
    - Appendix D now contains definitions of new kernel families and results for those kernels, as well as a discussion on why the bandwidth affects variance collapse. In total nine kernel variations are compared to SVGD.
    - The advantages of h-SVGD over other methods for avoiding variance underestimation such as GSVGD and S-SVGD (essentially, lower computational cost) are now more clearly discussed in Section 5.
    Note that we do not implement these competitor methods as the primary aim of this paper is to provide theoretical justification of h-SVGD, and we feel that such comparisons are more relevant for research that makes algorithmic contributions in this field.

* **h-KSD and the descent lemma:** Some reviewers pointed to the discussion in Appendix C that h-KSD is not a valid discrepancy measure. We wish to emphasise that this does not prevent the results in Section 4 from providing a solid theoretical underpinning of h-SVGD. In particular, the descent lemma (Theorem 4.2) provides bounds in terms of a genuine discrepancy measure (namely the KSD of one of the two kernels) so as to ensure the approximating distribution approaches the target distribution. To this end, we have made the following changes:

    - Further discussion has been added to Section 4.3, including a second upper bound on the step size in order to highlight that the KL divergence will decrease for a sufficiently small step size.
    - The condition on the step size in the statement of Theorem 4.2 has been relaxed to an inequality for better readability in light of the above point. No changes to the proof are required.
    - Remark 2 has been reworded and moved from Section 4.3 to Section 4.2 since it relates to Theorem 4.1 as well as Theorem 4.2.

Please also see individualised responses to each Reviewer's comments.

---

### Meta-Review · Area_Chair_SuK6 · 2023-12-06

**Metareview:**

This paper proposed a theoretical framework for using hybrid kernels in Stein variational gradient descent (SVGD). The idea is to use two different kernels for the drift and the repulsive terms in SVGD. The main contribution is on the theoretical analyses, e.g., in descending KL divergence and large time/particle asymptotic behaviours. Experiments on sampling high-dimensional Gaussians demonstrated improvements, although the improvements are less significant on Bayesian neural network regression benchmarks.

The two main issues raised by the reviewers are:
1. Not enough empirical support for the advantages of h-SVGD over S-SVGD and G-SVGD, especially in predictive performance when applied to e.g., Bayesian neural networks.
2. Proof of convergence is not clear, especially regarding whether the magnitude of the descending direction (h-KSD) is a valid discrepancy or not.

In revision the authors tried to clarify the proof, but questions still remain from the reviewers. Note that a paper needs to be solid on  at least one of the aspects (theory & empirical algorithmic performance) in order to be accepted. Given that the current version of this submission still has rooms to improve on both aspects, I recommend another round of revision-then-resubmit for this paper.

**Justification For Why Not Higher Score:**

The current version of this submission still has rooms to improve on both theoretical and practical aspects.

**Justification For Why Not Lower Score:**

N/A

---

### Decision · Program_Chairs · 2024-01-16

Reject